# FEW-SHOT ADVERSARIAL LOW-RANK FINE-TUNING OF VISION-LANGUAGE MODELS

## ABSTRACT

Vision-Language Models (VLMs) such as CLIP have shown remarkable performance in cross-modal tasks through large-scale contrastive pre-training. To adapt these large transformer-based models efficiently for downstream tasks, Parameter-Efficient Fine-Tuning (PEFT) techniques like (Low-Rank Adaptation) LoRA have emerged as scalable alternatives to full fine-tuning, especially in few-shot scenarios. However, like traditional deep neural networks, VLMs are highly vulnerable to adversarial attacks, where imperceptible perturbations can significantly degrade model performance. Adversarial training remains the most effective strategy for improving model robustness in PEFT. In this work, we propose AdvCLIP-LoRA, to our knowledge the first method designed to enhance the adversarial robustness of CLIP models fine-tuned with LoRA in few-shot settings. Our method formulates training as a minimax optimization over low-rank adapters and adversarial perturbations, enabling robust adaptation with a small trainable footprint. Across eight datasets and two backbones (ViT-B/16 and ViT-B/32), AdvCLIP-LoRA achieves state-of-the-art performance in few-shot classification, adversarial base-to-new generalization, and cross-dataset transfer, delivering higher adversarial robustness than prompt tuning baselines without sacrificing much clean accuracy. These findings highlight AdvCLIP-LoRA as a practical approach for robust adaptation of VLMs in resource-constrained settings.

## 1 INTRODUCTION

Vision-Language Models (VLMs), such as CLIP Radford et al. (2021), have become foundational in learning cross-modal representations by aligning visual and textual embeddings through large-scale contrastive pre-training Jia et al. (2021); Li et al. (2022b); Yao et al.. While these models enable effective zero-shot and few-shot adaptation Zhang et al. (2022); Zhu et al. (2023), their larger transformer-based variants Vaswani (2017) demonstrate superior performance (e.g., CLIP's ViT-L/14 surpasses ViT-B/16 by over 6% on ImageNet Deng et al. (2009)). However, these large models typically contain billions of trainable parameters, making full fine-tuning (FFT) computationally expensive and inefficient, particularly for task-specific adaptations.

To address this, Parameter-Efficient Fine-Tuning (PEFT) methods have gained traction, particularly in NLP, where techniques like adapters Chen et al. (2022); Karimi Mahabadi et al. (2021); Rebuffi et al. (2017) and prompt tuning Jia et al. (2022); Li & Liang (2021) reduce overhead, by adding a small number of trainable parameters or trainable prompt tokens while keeping the rest of the model frozen. Among PEFT methods, Low-Rank Adaptation (LoRA) Hu et al. (2021) offers an efficient alternative by fine-tuning only low-rank matrices, enabling single-GPU adaptation of billion-parameter models Dettmers et al. (2023) while matching full fine-tuning performance Hu et al. (2021). Recent work by Zanella & Ben Ayed (2024) employed LoRA in the context of few-shot VLMs, demonstrating improved accuracy across various tasks and models. Unlike few-shot prompt tuning Bulat & Tzimiropoulos (2023); Chen et al.; Zhu et al. (2023), which involves computationally intensive optimization of textual prompts, or adapter-based methods Gao et al. (2024); Zhang et al. (2022) that often demand extensive hyperparameter tuning Silva-Rodriguez et al. (2024), LoRA provides a more scalable and portable solution for fine-tuning VLMs Zanella & Ben Ayed (2024).

Despite their impressive capabilities, VLMs share the susceptibility of traditional deep neural networks (DNNs) to adversarial attacks, where imperceptible perturbations can significantly degrade

model performance Szegedy et al. (2013); Zhou et al. (2023). This vulnerability is particularly concerning in the visual domain, where adversarial noise can be more subtle and difficult to detect compared to textual modifications. Extensive research in computer vision has demonstrated that adversarial training remains the most effective approach for developing robust DNNs resistant to adversarial perturbations Madry et al. (2018). When applied to PEFT paradigms, this adversarial training is typically implemented during the fine-tuning phase rather than during initial pre-training. More recently, studies Li et al. (2024); Zhang et al. (2024); Jia et al. (2025) have explored few-shot prompt tuning as a means of adversarial adaptation. For instance, Zhang et al. (2024) trains the clean text embedding with the adversarial image embedding to improve adversarial robustness. APT Li et al. (2024) learns robust text prompts via adversarial training, while FAP Zhou et al. (2023) leverages multimodal prompts and proposes a loss function that balances the connection between natural and adversarial features across modalities.

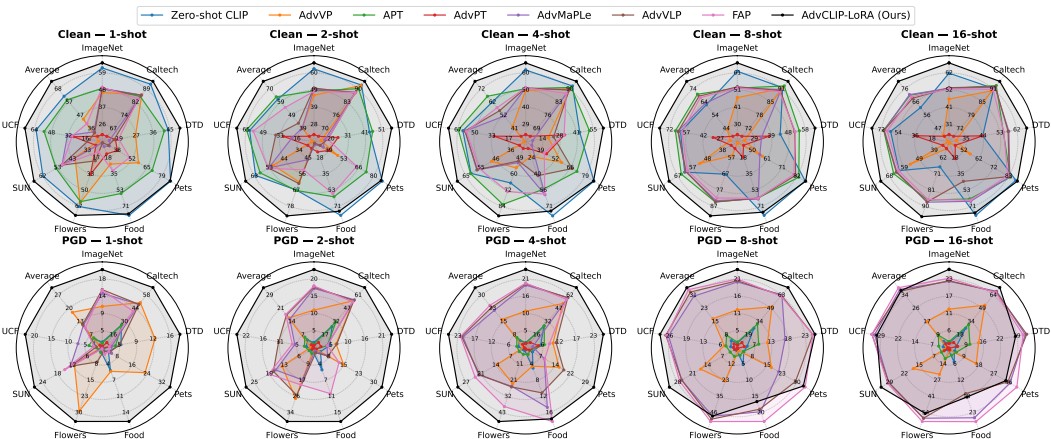

Figure 1: **Few-shot performance across datasets under clean and adversarial evaluation.** Spider plots show top-1 accuracy (%) for Clean (top row) and PGD-100 (bottom row) on eight datasets at shot counts $\{1, 2, 4, 8, 16\}$ with ViT-B/32. Each polygon denotes a method (larger area is better).

Despite their effectiveness, adversarial prompt-based methods exhibit two limitations: **(i)** they often attain robustness by sacrificing substantial clean accuracy, especially in the extreme few-shot regime (1–4 shots), where many underperform even zero-shot CLIP (Fig. 1, top); and **(ii)** their robustness typically improves only as the shot count increases, with some methods struggling to gain robustness in the extreme few-shot regime (Fig. 1, bottom). Although LoRA has proven effective for standard fine-tuning, its use for enhancing adversarial robustness in *few-shot VLMs* remains largely unexplored. We address this gap with AdvCLIP-LoRA, which fine-tunes CLIP using LoRA adapters under a minimax objective. As shown in Fig. 1, our simple AdvCLIP-LoRA avoids the above trade-offs, delivering superior robustness *and* higher clean accuracy, consistently outperforming adversarial prompt-tuning baselines on both clean and PGD metrics for the majority of shots.

Before delving into the details, we summarize our main contributions.

- We investigate LoRA for adversarially robust few-shot VLMs, a setting largely dominated by prompt-based strategies, and introduce AdvCLIP-LoRA, which frames adaptation as a minimax optimization problem and solves it efficiently.

- We conduct extensive experiments on eight datasets with ViT-B/16 and ViT-B/32 backbones, covering few-shot classification, adversarial base-to-new generalization, and cross-dataset transfer; AdvCLIP-LoRA significantly improves robustness to strong attacks (e.g., PGD) in most settings with minimal loss in clean accuracy.

- We present comprehensive ablations that analyze design choices and hyperparameters, providing guidance for practical deployment.

- Under standard assumptions from the minimax optimization literature (e.g., smooth objectives and bounded gradients), we establish convergence guarantees for the primal function $\Phi(\cdot) = \max_{\delta \in \Delta} f(\cdot, \delta)$ to a stationary point, with rates matching classical results.

## 2 PRELIMINARIES AND RELATED WORK

### 2.1 FEW-SHOT FINE-TUNING FOR VLMS

In vision-language classification tasks, predictions are made by leveraging the pretrained alignment between visual and textual modalities. Given a label set of $K$ classes, one first constructs natural language descriptions, or prompts Liu et al. (2023a), denoted as $\{c_k\}_{k=1}^K$, where each $c_k$ is a textual phrase such as "a photo of a [class name]." These prompts are embedded using a frozen text encoder $\theta_t$, yielding normalized representations $\mathbf{z}_k^{(T)} = \theta_t(c_k) \in \mathbb{R}^d$. Similarly, an image $\mathbf{x}_i$ is embedded via a visual encoder $\theta_v$ to obtain $\mathbf{z}_i^{(I)} = \theta_v(\mathbf{x}_i) \in \mathbb{R}^d$, also normalized to unit length. The prediction logits are computed as the cosine similarity between each image-text pair. These logits are converted into a probability distribution over classes using a softmax with temperature scaling:

$$p_{i,k} = \frac{\exp(\cos(\mathbf{z}_i^{(I)}, \mathbf{z}_k^{(T)})/\gamma)}{\sum_{j=1}^K \exp(\cos(\mathbf{z}_i^{(I)}, \mathbf{z}_j^{(T)})/\gamma)}, \tag{1}$$

where $\gamma$ is a softmax-temperature parameter. The predicted label for image $\mathbf{x}_i$ is the one with the highest posterior probability: $\hat{k} = \arg\max_k p_{i,k}$. This form of zero-shot prediction directly mirrors the contrastive training setup used in large-scale VLM pretraining, such as CLIP Radford et al. (2021), and allows models to generalize to novel classification tasks without fine-tuning on the target domain.

To further adapt vision-language models to downstream tasks, the few-shot setting assumes access to a limited number of labeled examples per target class—typically fewer than 16. Given $N$ such labeled support images per class, we denote the one-hot encoded ground-truth label for image $\mathbf{x}_i$ as $y_{ik}$, where $y_{ik} = 1$ if $\mathbf{x}_i$ belongs to class $k$, and 0 otherwise. Classification probabilities $p_{i,k}$ are obtained as in the zero-shot setup, and the model is adapted by minimizing the cross-entropy loss:

$$\mathcal{L}_{\text{CE}} = -\frac{1}{N} \sum_{i=1}^N \sum_{k=1}^K y_{ik} \ln p_{i,k}. \tag{2}$$

This adaptation can be implemented in several ways. One strategy is to optimize the input prompts $\{c_k\}_{k=1}^K$ directly, an approach inspired by prompt tuning techniques Chen et al.. Alternatively, one may fine-tune lightweight, task-specific modules such as adapter layers Gao et al. (2024) or low-rank parameterizations like LoRA Zanella & Ben Ayed (2024), leaving the backbone encoders frozen.

### 2.2 FINE-TUNING VLMS VIA LORA

Low-Rank Adaptation (LoRA) Hu et al. (2021) is a highly promising PEFT method, enabling efficient fine-tuning of large models by freezing the entire pre-trained model and introducing low-rank, trainable matrices within each layer. In LoRA, given a pre-trained weight matrix $W_0 \in \mathbb{R}^{d \times k}$, the weight update is achieved through a low-rank decomposition $W_0 + \Delta W = W_0 + BA$, where the training occurs on matrices $A \in \mathbb{R}^{r \times k}$ and $B \in \mathbb{R}^{d \times r}$, with $r \ll \min(d, k)$. The values in $A$ are initialized via a Gaussian distribution, while $B$ is initialized as a zero matrix. This setup ensures that no low-rank update occurs before training, meaning that the output remains unchanged initially.

Although the original LoRA paper applies the low-rank matrices to the attention matrices of transformer-based architectures, Zanella & Ben Ayed (2024) extends LoRA to all matrices in the vision and text encoders of VLMs. This adaptation leads to improved accuracy over prompt-based methods across various CLIP architectures and datasets Zanella & Ben Ayed (2024).

### 2.3 ADVERSARIAL ROBUSTNESS

Given an arbitrary classifier $h : \mathcal{X} \to \mathcal{Y}$, where an input $x \in \mathcal{X}$ is associated with its true label $y \in \mathcal{Y}$, an adversary attempts to find an imperceptible perturbation $\delta$, which shares the same dimensionality as $x$. This perturbation must satisfy the condition that $x + \delta \in \mathcal{X}$, and more critically, $h(x + \delta) \neq y$, thereby misclassifying the original input. To ensure that this perturbation remains imperceptible, the adversarial perturbation $\delta$ is usually constrained within some bounded set $\Delta \subseteq \mathbb{R}^n$.

The adversarial attack on a classifier $h$, constrained by bounded set $\Delta$, is formulated as follows:

$$\hat{x} = x + \arg\max_{\delta \in \Delta} \mathcal{L}(h(x + \delta), y), \tag{3}$$

Figure 2: 🔥: Trainable Parameters, ❄: Frozen Parameters. Illustration of AdvCLIP-LoRA algorithm. During iteration $t$, the perturbation $\delta_t$ is updated and applied to the input image batch. Subsequently, the low-rank matrices $A$ and $B$ are optimized, while the rest of the model remains frozen.

---

**Algorithm 1** AdvCLIP-LoRA

---

**Require:** Training samples $\mathcal{X}$, batch-size $M$, learning rates $\eta_w, \eta_\delta$
1:  $A_0 \sim \mathcal{N}(0, \sigma^2)$, $B_0 = 0$.
2:  $\delta \leftarrow 0$
3:  **for** epoch $= 1 \ldots T$ **do**
4:      **for** minibatch $M \subset \mathcal{X}$ **do**
5:          **for** $j = 1 \ldots \tau$ **do**
6:              $\delta_t = \mathcal{P}_\Delta \left( \delta_{t-1} + \eta_\delta \left( \frac{1}{M} \sum_{i=1}^M \nabla_\delta F(W_{t-1}, \delta_{t-1}; \xi_i) \right) \right)$
7:          **end for**
8:          $A_t = A_{t-1} - \eta_w \left( \frac{1}{M} \sum_{i=1}^M \nabla_A F(W_{t-1}, \delta_t; \xi_i) \right)$    ▷ Update the low-rank matrix $A$
9:          $B_t = B_{t-1} - \eta_w \left( \frac{1}{M} \sum_{i=1}^M \nabla_B F(W_{t-1}, \delta_t; \xi_i) \right)$    ▷ Update the low-rank matrix $B$
10:     **end for**
11: **end for**

---

where $\mathcal{L}$ is the training loss function. This formulation represents an optimization problem where the perturbation $\delta$ is chosen such that the classifier's output is maximally disrupted while staying within a bounded set. Methods like Projected Gradient Descent (PGD) Madry et al. (2018) are commonly employed to solve this optimization problem. Given the vulnerability of deep learning models to these perturbations Szegedy et al. (2013), it becomes crucial to defend against such adversarial attacks.

One of the most effective strategies for defending against adversarial attacks is adversarial training, as proposed by Madry et al. (2018). When $h_W$ denotes a classifier parameterized by $W$, adversarial training seeks to solve the following minimax optimization problem:

$$\min_W \mathbb{E}_{(x,y) \sim \mathcal{D}} \left[ \max_{\delta \in \Delta} \mathcal{L}(h_W(x + \delta), y) \right], \tag{4}$$

where $\mathcal{D}$ represents the underlying data distribution. This approach effectively trains the classifier to be robust against adversarial perturbations by simultaneously minimizing the classifier's loss and maximizing perturbation within a bounded set.

## 3    PROPOSED ALGORITHM

### 3.1    ADVERSARIAL FINE-TUNING OF CLIP VIA LORA

Assume that the LoRA matrices $A$ and $B$ are initialized with a Gaussian distribution and zero matrices, respectively, and are applied to all weight matrices in the vision and text encoders of a CLIP model. Following the approach introduced in Section 2.3, we aim to improve the adversarial robustness of the LoRA-based CLIP model by introducing a perturbation $\delta$ to input images and solving a minimax optimization problem. Focusing on the dependence of the training loss function on the low-rank matrices $A$ and $B$ and the perturbation $\delta$, we formulate the following minimax optimization problem:

$$\min_{A,B} \max_{\delta \in \Delta} f(W := W_0 + BA, \delta), \tag{5}$$

where $\Delta$ is a bounded set of admissible perturbations, and $f : \mathbb{R}^{d \times k + n} \to \mathbb{R}$ is a non-convex loss function expressible in the stochastic form $\mathbb{E}_{\xi \sim \mathcal{D}}[F(W_0 + BA, \delta; \xi)]$. Here, the expectation is taken over sampled batches $\xi \sim \mathcal{D}$, where $\mathcal{D}$ represents the underlying data distribution.

## 3.2 AdvCLIP-LoRA Algorithm

In this section, we present the proposed AdvCLIP-LoRA algorithm, which solves the minimax problem (Eq. 5) to enhance the adversarial robustness of a CLIP model fine-tuned with LoRA. The AdvCLIP-LoRA algorithm proceeds for $T$ iterations. At each iteration $t$:

1) Select $M$ samples $\{\xi_i\}_{i=1}^M$ from the dataset.

2) Update the perturbation $\delta$ for $\tau$ iterations via:

$$\delta_t = \mathcal{P}_\Delta(\delta_{t-1} + \frac{\eta_\delta}{M} \sum_{i=1}^M \nabla_\delta F(W_{t-1}, \delta_{t-1}; \xi_i)), \qquad (6)$$

where $\eta_\delta$ is the learning rate for $\delta$, $\Delta$ is a bounded perturbation set, and $\mathcal{P}_\Delta$ projects onto $\Delta$. The set $\Delta$ may be any convex, bounded subset of $\mathbb{R}^n$; in our experiments we take $\Delta = \{\delta : \|\delta\|_\infty \le \epsilon\}$, i.e., an $\ell_\infty$-ball of radius $\epsilon$.

3) Update the LoRA matrices $A$ and $B$ using the current $\delta_t$ to obtain $A_t$ and $B_t$ (lines 8 and 9 of Alg. 1), where $\eta_w$ is the learning rate for $A$ and $B$. The steps of the AdvCLIP-LoRA algorithm are illustrated in Fig. 2. Moreover, the AdvCLIP-LoRA pipeline can be found in Alg. 1.

## 4 Convergence Analysis

In this section, we present a thorough convergence analysis of the proposed AdvCLIP-LoRA algorithm. The complete proofs can be found in Appendix C.

Consider the minimax problem (Eq. 5), which is equivalent to minimizing the function $\Phi(\cdot) = \max_{\delta \in \Delta} f(\cdot, \delta)$. In the context of nonconvex-strongly-concave minimax problems, where $f(W, \cdot)$ is strongly-concave for each $W$, the maximization problem $\max_{\delta \in \Delta} f(W, \delta)$ can be solved efficiently, yielding useful insights into $\Phi$. However, finding the global minimum of $\Phi$ remains NP-hard in general due to its nonconvex nature. To address this challenge, we define local surrogates for the global minimum of $\Phi$. One commonly used surrogate in nonconvex optimization is the notion of *stationarity*, which is suitable when $\Phi$ is differentiable. A point $W$ is an $\epsilon$-stationary point ($\epsilon \ge 0$) of a differentiable function $\Phi$ if $\|\nabla\Phi(W)\| \le \epsilon$.

Let us proceed with a few assumptions. Note that $\|\cdot\|_F$ denotes the Frobenius norm.

**Assumption 4.1** *We assume that the stochastic gradients are unbiased and bounded, that is,*

$$\mathbb{E}_\xi [\nabla F (W, \delta; \xi)] = \nabla f (W, \delta), \quad \mathbb{E}_\xi \left[\|\nabla F (W, \delta; \xi)\|_F^2\right] \le G^2, \qquad (7)$$

*for all $W \in \mathbb{R}^{d \times k}$, where $\xi$ represents a randomly sampled subset of training data and $\mathbb{E}_\xi[\cdot]$ denotes the expectation over $\xi \sim \mathcal{D}$.*

**Assumption 4.2** *The objective function and constraint set $\left(f : \mathbb{R}^{d \times k + n} \to \mathbb{R}, \Delta \subseteq \mathbb{R}^n\right)$ satisfy (i) $\Delta$ is a convex and bounded set with a diameter $D \ge 0$. (ii) $f$ has $\ell$-Lipchits gradients and is $\mu$-strongly concave in $\delta$. That is, for both $* \in \{W, \delta\}$*

$$\|\nabla_* f(W, \delta) - \nabla_* f (W', \delta')\|_F^2 \le \ell^2 \left(\|W - W'\|_F^2 + \|\delta - \delta'\|_F^2\right). \qquad (8)$$

Let $\kappa = \ell/\mu$ denote the condition number and define

$$\Phi(\cdot) = \max_{\delta \in \Delta} f(\cdot, \delta), \quad \delta^\star(\cdot) = \operatorname*{argmax}_{\delta \in \Delta} f(\cdot, \delta). \qquad (9)$$

The following theorem characterizes the convergence rate of the proposed AdvCLIP-LoRA in Alg. 1 to find a stationary solution for $\Phi(W)$.

Table 1: Few-shot classification under clean and adversarial evaluation (1-, 4-, and 16-shot).

| Shots | Method | ImageNet-1K | | Caltech101 | | DTD | | OxfordPets | | Food101 | | Flowers102 | | SUN397 | | UCF101 | | Average | |
|---|---|---|---|---|---|---|---|---|---|---|---|---|---|---|---|---|---|---|---|
| | | Clean | PGD | Clean | PGD | Clean | PGD | Clean | PGD | Clean | PGD | Clean | PGD | Clean | PGD | Clean | PGD | Clean | PGD |
| 1 | AdvVP Mao et al. | 46.60 | 11.07 | 85.73 | 50.33 | 26.97 | 12.93 | 24.43 | 5.23 | 57.60 | 22.73 | 63.10 | 29.70 | 3.37 | 0.40 | 41.20 | 11.10 | 43.62 | 17.94 |
| | APT Li et al. (2024) | 49.30 | 1.30 | 84.77 | 26.90 | 41.67 | 3.83 | 56.57 | 0.83 | 70.23 | 0.60 | 61.97 | 2.10 | 54.50 | 3.87 | 53.53 | 1.23 | 59.07 | 5.08 |
| | AdvPT Zhang et al. (2024) | 20.17 | 0.43 | 62.97 | 7.60 | 16.73 | 2.60 | 13.27 | 0.00 | 37.93 | 0.13 | 33.97 | 0.43 | 27.03 | 0.00 | 27.57 | 0.37 | 29.96 | 1.44 |
| | AdvMaPLe Khattak et al. (2023) | 49.27 | 14.60 | 85.53 | 48.37 | 13.63 | 2.93 | 5.27 | 0.30 | 30.67 | 4.97 | 1.40 | 0.10 | 32.70 | 7.07 | 49.70 | 12.67 | 33.52 | 11.38 |
| | AdvVLP Zhou et al. (2024) | 50.53 | 17.50 | 85.43 | 48.47 | 15.97 | 4.77 | 1.07 | 0.77 | 29.63 | 3.83 | 19.77 | 6.57 | 11.83 | 1.73 | 49.83 | 12.60 | 33.01 | 12.03 |
| | FAP Zhou et al. (2024) | 49.90 | 15.40 | 83.53 | 41.13 | 18.40 | 2.40 | 31.67 | 1.43 | 49.23 | 3.47 | 10.40 | 0.53 | 28.50 | 2.43 | 49.53 | 14.93 | 40.14 | 10.22 |
| | AdvCLIP-LoRA | 65.28 | 20.89 | 93.06 | 66.49 | 49.65 | 18.68 | 78.90 | 16.41 | 86.97 | 36.03 | 76.17 | 34.55 | 72.48 | 22.34 | 67.92 | 26.99 | 73.80 | 30.30 |
| | *Relative Improvement* | +29.19 | +19.37 | +8.55 | +32.11 | +19.15 | +44.47 | +39.47 | +163.84 | +23.84 | +58.51 | +20.71 | +16.33 | +32.99 | +163.84 | +26.88 | +80.78 | +24.94 | +68.9 |
| 4 | AdvVP Mao et al. | 49.80 | 11.13 | 90.17 | 52.50 | 18.77 | 9.27 | 22.73 | 4.57 | 57.80 | 16.20 | 55.97 | 23.73 | 1.07 | 0.80 | 48.47 | 13.03 | 43.10 | 16.40 |
| | APT Li et al. (2024) | 50.90 | 1.40 | 90.77 | 26.67 | 51.33 | 6.33 | 54.80 | 1.63 | 71.83 | 2.10 | 82.40 | 4.23 | 66.53 | 3.03 | 62.37 | 2.90 | 66.37 | 6.04 |
| | AdvPT Zhang et al. (2024) | 23.40 | 1.33 | 64.97 | 7.30 | 31.70 | 4.37 | 15.23 | 0.37 | 44.13 | 1.73 | 41.97 | 0.63 | 31.17 | 0.47 | 29.97 | 0.40 | 35.32 | 2.07 |
| | AdvMaPLe Khattak et al. (2023) | 51.27 | 19.00 | 89.53 | 59.40 | 6.43 | 2.40 | 60.00 | 14.83 | 30.70 | 9.03 | 52.20 | 25.37 | 59.73 | 21.30 | 58.23 | 21.53 | 51.01 | 21.61 |
| | AdvVLP Zhou et al. (2024) | 51.30 | 19.37 | 89.37 | 59.07 | 22.97 | 10.33 | 41.50 | 11.20 | 67.43 | 18.47 | 51.00 | 25.80 | 59.97 | 21.77 | 57.90 | 21.17 | 55.18 | 23.40 |
| | FAP Zhou et al. (2024) | 51.53 | 19.60 | 87.57 | 57.33 | 31.27 | 8.07 | 59.37 | 23.20 | 79.47 | 34.57 | 81.53 | 52.63 | 60.70 | 26.67 | 60.40 | 27.23 | 57.51 | 24.60 |
| | AdvCLIP-LoRA | 66.34 | 23.78 | 93.96 | 71.03 | 62.41 | 26.36 | 75.80 | 17.69 | 87.03 | 32.98 | 90.70 | 48.72 | 76.18 | 26.22 | 71.09 | 31.11 | 77.94 | 34.74 |
| | *Relative Improvement* | +28.74 | +21.33 | +3.51 | +19.58 | +21.59 | +155.18 | +27.67 | −23.75 | +9.51 | −4.6 | +10.07 | −7.43 | +14.5 | −1.69 | +13.98 | +14.25 | +17.43 | +41.22 |
| 16 | AdvVP Mao et al. | 46.27 | 12.77 | 90.40 | 52.60 | 29.20 | 13.87 | 1.07 | 0.80 | 56.40 | 16.43 | 56.17 | 22.03 | 0.97 | 0.93 | 54.70 | 17.63 | 41.90 | 17.13 |
| | APT Li et al. (2024) | 52.63 | 2.07 | 92.93 | 30.23 | 54.93 | 10.47 | 62.50 | 2.63 | 83.70 | 4.40 | 86.63 | 8.97 | 69.40 | 4.40 | 65.67 | 3.67 | 71.05 | 8.35 |
| | AdvPT Zhang et al. (2024) | 24.53 | 1.47 | 68.70 | 9.63 | 43.77 | 5.70 | 18.47 | 0.73 | 46.27 | 0.23 | 56.03 | 0.80 | 36.60 | 0.53 | 33.13 | 2.37 | 40.94 | 2.68 |
| | AdvMaPLe Khattak et al. (2023) | 52.93 | 21.90 | 92.17 | 68.65 | 57.93 | 32.17 | 65.13 | 25.27 | 83.27 | 36.87 | 87.87 | 58.70 | 68.97 | 31.67 | 63.57 | 29.70 | 71.48 | 38.11 |
| | AdvVLP Zhou et al. (2024) | 53.23 | 22.10 | 92.37 | 67.97 | 57.53 | 32.73 | 43.30 | 16.50 | 82.93 | 35.57 | 87.70 | 58.70 | 69.10 | 32.80 | 63.90 | 29.70 | 68.76 | 37.01 |
| | FAP Zhou et al. (2024) | 52.53 | 22.90 | 91.10 | 67.33 | 55.17 | 31.33 | 64.03 | 26.67 | 81.90 | 41.00 | 86.27 | 61.47 | 65.70 | 32.80 | 62.37 | 30.27 | 69.88 | 39.22 |
| | AdvCLIP-LoRA | 68.38 | 25.86 | 94.93 | 72.98 | 67.67 | 28.37 | 77.81 | 17.76 | 88.44 | 34.29 | 96.47 | 54.69 | 81.87 | 30.74 | 74.23 | 33.52 | 81.23 | 37.28 |
| | *Relative Improvement* | +28.46 | +12.93 | +2.15 | +6.34 | +16.81 | −13.32 | +19.47 | −33.41 | +5.66 | −16.37 | +9.79 | −11.03 | +17.97 | −6.28 | +13.03 | +10.74 | +13.64 | −4.95 |

**Theorem 4.1** *Let Assumptions 4.1 and 4.2 hold. Moreover, assume that the low-rank matrices remain bounded by constants $c_A$ and $c_B$ in each iteration, i.e., $\|A_t\|_F \leq c_A$ and $\|B_t\|_F \leq c_B$. Then, there exists iteration $t \in \{0, \cdots, T-1\}$ for which*

$$\mathbb{E}\left\|\nabla\Phi\left(W_t\right)\right\|_F^2 \leq \mathcal{O}\left(\frac{4\Delta_\Phi(1/\eta_w) + \kappa\ell^2(c_A^2 + c_B^2)D^2}{\epsilon^2}\right), \tag{10}$$

*where $\eta_w = \Theta(\min\{1/\kappa\ell(c_A^4 + c_B^4), 1/\kappa^2\ell(c_A^2 + c_B^2), 1/(G^2 + \kappa\ell c_A^4 c_B^2)^{1/2}\})$, $\eta_\delta = \Theta(1/\ell)$, and $\Delta_\Phi = \mathbb{E}\Phi(W_0) - \mathbb{E}\Phi(W_{T+1})$. Moreover, the mini-batch size $M$ is bounded by*

$$\mathcal{O}\left(\frac{G^2 + \kappa(c_A^2 + c_B^2)G^2}{\epsilon^2}\right). \tag{11}$$

**Remark 4.1** *AdvCLIP-LoRA is guaranteed to reach an $\epsilon$-stationary point of $\Phi(\cdot)$ in $\mathcal{O}(\epsilon^{-2})$ iterations, with total stochastic gradient complexity $\mathcal{O}(\epsilon^{-4})$, matching classical rates in the minimax optimization literature Lin et al. (2020).*

## 5 EMPIRICAL RESULTS

### 5.1 EXPERIMENTAL SETUP

**Datasets.** To evaluate the proposed method, we follow prior works Zhou et al. (2022); Jia et al. (2025) and utilize a diverse set of 8 image recognition datasets spanning multiple vision tasks. The datasets include two generic object recognition datasets: ImageNet-1K Deng et al. (2009) and Caltech101 Fei-Fei et al. (2004); a texture recognition dataset: DTD Cimpoi et al. (2014); four fine-grained object recognition datasets: OxfordPets Parkhi et al. (2012), Flowers102 Nilsback & Zisserman (2008), and Food101 Bossard et al. (2014); a scene recognition dataset: SUN397 Xiao et al. (2010); and an action recognition dataset: UCF101 Soomro et al. (2012).

**Baselines.** To rigorously evaluate the proposed method, we benchmark it against a representative set of adversarial prompt-learning baselines. We consider two categories: (i) methods using hand-crafted text supervision, such as zero-shot CLIP Radford et al. (2021) and AdvVP Mao et al.; and (ii) methods with learnable text prompts. In the single-modality textual setting, we compare against APT Li et al. (2024), which learns robust text prompts without modifying model parameters, and AdvPT Zhang et al. (2024), which first employs the image encoder to generate adversarial examples and then aligns them with learnable text prompts. For multimodal adversarial prompt learning, we follow Zhou et al. (2024) and include AdvVLP, AdvMaPLe Khattak et al. (2023), and FAP Zhou et al. (2024).

**Implementation Details.** We conduct experiments with CLIP backbones ViT-B/16 and ViT-B/32 and report averages over three random seeds. The base optimizer uses a learning rate of $2 \times 10^{-4}$ with a cosine decay schedule. Learning the perturbation $\delta$ is challenging early in training due to small gradients; to mitigate this, we employ a larger, adaptive rate $\eta_\delta = 0.05/\|\delta_t\|_2$, which scales inversely with the current perturbation magnitude. This choice amplifies early updates and serves as implicit data augmentation by injecting noise. $\eta_\delta$ then decays during training and is fixed at 0.05

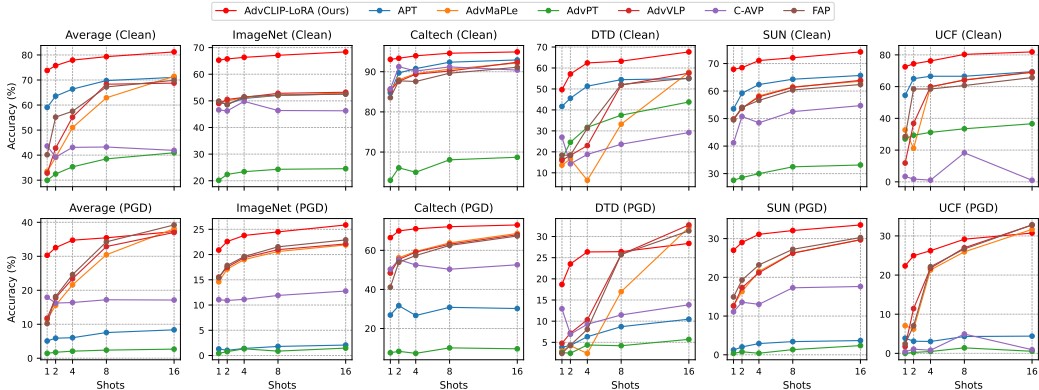

Figure 3: **Effect of shot count on clean and adversarial performance.** Clean and PGD accuracy versus number of shots $\{1, 2, 4, 8, 16\}$ on representative datasets and the eight-dataset average.

from iteration 300 onward. The total number of training iterations is $500 \times N/K$. We use a batch size of 16 for ImageNet-1K and 32 for all other datasets.

For LoRA, the class-conditional prompt is "a photo of a `kth class name`," $k = 1, \ldots, K$, to demonstrate AdvCLIP-LoRA's applicability without elaborate manual prompt engineering. LoRA modules are inserted at all layers of both encoders with rank 2 and dropout $p = 0.25$. Attacks are generated within an $\ell_\infty$-ball using a 2-step PGD procedure with budget $\epsilon = 1/255$ and step size $\alpha = 1/255$, following Mao et al.; robustness is evaluated with a 100-step PGD attack. All experiments are run on NVIDIA A6000 and V100 GPUs.

## 5.2 PERFORMANCE EVALUATION

**Adversarial Few-Shot Learning.** We assess performance under scarce supervision by fine-tuning with $\{1, 2, 4, 8, 16\}$ shots per class. Table 1 reports results for the 1-, 4-, and 16-shot settings across eight datasets; results for the remaining shot counts are provided in the Appendix. We also report the *relative improvement* of AdvCLIP-LoRA over the strongest non-ours baseline for each setting. Overall, AdvCLIP-LoRA consistently delivers higher clean accuracy with substantial margins. Under PGD evaluation, the advantage is pronounced at 1–4 shots, remains favorable at 8 shots, and narrows at 16 shots, where performance is slightly trailing the best baseline (FAP). Fig. 3 visualizes clean and PGD accuracy as a function of shots for representative datasets and the eight-dataset average, *highlighting that while some prompt-based baselines improve as shots increase, others fail to improve*, whereas AdvCLIP-LoRA is already strong from the 1-shot regime.

**Adversarial Base-to-New Generalization.** We present a more challenging adversarial base-to-new generalization setting in which each dataset is partitioned into base and new subclasses. Models are fine-tuned with 16 shots per base class and then evaluated on both base and new classes under clean and PGD-100 conditions. As the number of categories is typically modest relative to the per-class sample count, this setting requires learning intrinsic, dataset-level structure and robust representations from limited supervision that transfer to a large test pool. Table 2 presents results together with *relative improvement*. AdvCLIP-LoRA attains consistently superior clean and adversarial accuracy on both base and new splits; moreover, the gains are larger on the new classes, highlighting stronger robustness and generalization to previously unseen categories.

**Adversarial Cross-Dataset Evaluation.** We assess zero-shot transfer robustness via cross-dataset generalization. A CLIP backbone is first adversarially fine-tuned on ImageNet-1K with 16 shots per class, then evaluated without further fine-tuning on seven downstream datasets under Clean and PGD-100 conditions. Table 3 reports the results and the *relative improvement* of AdvCLIP-LoRA over the strongest non-ours baseline (excluding zero-shot CLIP). As expected, zero-shot CLIP attains strong clean accuracy but offers minimal adversarial resistance. Adversarially adapted models typically sacrifice some clean accuracy for robustness; AdvCLIP-LoRA shows the smallest drop in clean

Table 2: **Adversarial base-to-new generalization (16-shot).** Top-1 accuracy (%) on base and new classes under clean and PGD-100 evaluation across eight datasets.

| Clean Acc (%) | ImageNet-1K | | Caltech101 | | DTD | | OxfordPets | | Food101 | | Flowers102 | | SUN397 | | UCF101 | | Average | |
|---|---|---|---|---|---|---|---|---|---|---|---|---|---|---|---|---|---|---|
| | Base | New | Base | New | Base | New | Base | New | Base | New | Base | New | Base | New | Base | New | Base | New |
| AdvVP Mao et al. | 49.87 | 44.80 | 92.83 | 88.83 | 23.27 | 13.23 | 32.57 | 32.30 | 2.27 | 2.20 | 50.43 | 45.23 | 60.20 | 62.20 | 1.77 | 2.47 | 31.68 | 30.39 |
| APT Li et al. (2024) | 24.73 | 25.43 | 67.63 | 43.83 | 14.17 | 19.43 | 9.47 | 2.73 | 2.97 | 8.10 | 2.07 | 3.47 | 13.10 | 11.17 | 14.73 | 17.37 | 18.21 | 13.99 |
| AdvPT Zhang et al. (2024) | 26.53 | 69.03 | 72.27 | 62.33 | 52.70 | 46.77 | 53.43 | 51.17 | 25.07 | 53.70 | 70.23 | 46.70 | 41.40 | 59.17 | 43.47 | 43.60 | 43.87 | 44.94 |
| AdvVLP Khattak et al. (2023) | 58.40 | 48.83 | 94.40 | 83.27 | 43.40 | 21.27 | 38.97 | 39.67 | 71.37 | 68.93 | 88.90 | 49.90 | 70.23 | 63.57 | 72.77 | 49.83 | 60.38 | 46.18 |
| AdvMaPLe Zhou et al. (2024) | 58.47 | 48.67 | 94.87 | 84.47 | 48.63 | 22.87 | 60.67 | 57.90 | 71.40 | 69.90 | 56.53 | 30.00 | 70.57 | 63.27 | 72.80 | 50.70 | 58.95 | 46.92 |
| FAP Zhou et al. (2024) | 58.10 | 47.83 | 79.15 | 76.53 | 69.17 | 35.17 | 87.37 | 72.13 | 72.37 | 68.20 | 89.30 | 45.67 | 68.47 | 61.47 | 70.37 | 47.10 | 70.52 | 49.58 |
| AdvCLIP-LoRA (Ours) | 72.21 | 56.72 | 97.48 | 91.05 | 78.94 | 52.90 | 91.28 | 87.75 | 81.75 | 79.61 | 96.01 | 54.82 | 79.05 | 70.48 | 82.57 | 62.30 | 84.91 | 69.45 |
| *Relative Improvement* | *+23.5* | *+16.16* | *+2.75* | *+7.79* | *+14.12* | *+13.11* | *+4.48* | *+21.66* | *+12.96* | *+13.89* | *+7.51* | *+17.39* | *+12.02* | *+11.4* | *+13.42* | *+22.88* | *+20.41* | *+40.08* |

| PGD-100 Acc (%) | ImageNet-1K | | Caltech101 | | DTD | | OxfordPets | | Food101 | | Flowers102 | | SUN397 | | UCF101 | | Average | |
|---|---|---|---|---|---|---|---|---|---|---|---|---|---|---|---|---|---|---|
| | Base | New | Base | New | Base | New | Base | New | Base | New | Base | New | Base | New | Base | New | Base | New |
| AdvVP Mao et al. | 12.27 | 12.27 | 57.17 | 49.13 | 10.03 | 7.20 | 12.27 | 13.37 | 1.27 | 1.00 | 24.63 | 15.77 | 18.50 | 21.10 | 1.73 | 1.43 | 14.43 | 13.36 |
| APT Li et al. (2024) | 9.83 | 5.90 | 15.97 | 9.97 | 8.87 | 3.60 | 0.33 | 0.00 | 0.47 | 1.93 | 0.13 | 0.03 | 0.67 | 2.23 | 2.03 | 5.33 | 3.80 | 3.07 |
| AdvPT Zhang et al. (2024) | 0.50 | 14.77 | 13.60 | 15.17 | 7.13 | 6.83 | 1.27 | 8.53 | 1.63 | 10.97 | 1.17 | 9.93 | 3.77 | 12.83 | 6.63 | 6.60 | 3.50 | 8.84 |
| AdvVLP Khattak et al. (2023) | 25.33 | 21.03 | 73.90 | 56.70 | 21.50 | 9.97 | 16.80 | 17.50 | 27.90 | 24.50 | 62.80 | 21.07 | 33.87 | 29.83 | 36.37 | 20.13 | 30.69 | 20.25 |
| AdvMaPLe Zhou et al. (2024) | 24.93 | 20.50 | 76.23 | 57.67 | 27.57 | 12.37 | 31.80 | 28.90 | 28.43 | 24.60 | 36.70 | 11.63 | 34.10 | 29.40 | 36.77 | 20.00 | 32.37 | 21.61 |
| FAP Zhou et al. (2024) | 25.83 | 21.57 | 74.20 | 50.00 | 41.63 | 19.77 | 34.13 | 26.07 | 27.57 | 24.20 | 65.50 | 18.10 | 34.63 | 30.77 | 36.63 | 18.30 | 38.05 | 21.86 |
| AdvCLIP-LoRA (Ours) | 25.58 | 22.40 | 79.15 | 65.61 | 41.90 | 31.16 | 45.19 | 49.38 | 23.54 | 23.50 | 57.26 | 29.43 | 39.80 | 37.02 | 32.52 | 19.15 | 43.12 | 34.71 |
| *Relative Improvement* | *−0.97* | *+9.27* | *+3.83* | *+13.77* | *+0.65* | *+57.61* | *+32.41* | *+70.87* | *−17.2* | *−2.89* | *−12.58* | *+153.05* | *+16.72* | *+20.31* | *−11.56* | *+4.64* | *+13.32* | *+58.78* |

Table 3: **Cross-dataset generalization (zero-shot transfer).** Models are adversarially fine-tuned on ImageNet-1K with 16 shots, then evaluated *without* further adaptation on seven downstream datasets.

| Method | ImageNet-1K | | Caltech101 | | DTD | | OxfordPets | | Food101 | | Flowers102 | | SUN397 | | UCF101 | | Average | |
|---|---|---|---|---|---|---|---|---|---|---|---|---|---|---|---|---|---|---|
| | Clean | PGD | Clean | PGD | Clean | PGD | Clean | PGD | Clean | PGD | Clean | PGD | Clean | PGD | Clean | PGD | Clean | PGD |
| Zero-shot CLIP Radford et al. (2021) | 62.10 | 1.57 | 91.50 | 26.23 | 43.70 | 5.07 | 87.40 | 3.27 | 80.50 | 5.03 | 66.90 | 1.73 | 62.10 | 1.20 | 62.00 | 2.47 | 69.53 | 5.82 |
| AdvVP Mao et al. | 44.87 | 11.67 | 85.47 | 48.07 | 30.23 | 12.93 | 74.20 | 19.03 | 56.53 | 9.70 | 43.17 | 16.20 | 41.97 | 12.77 | 44.60 | 10.47 | 52.63 | 17.60 |
| APT Li et al. (2024) | 12.23 | 0.90 | 53.57 | 7.70 | 11.93 | 3.47 | 13.97 | 1.10 | 7.30 | 0.10 | 13.73 | 0.67 | 14.73 | 2.37 | 18.30 | 0.33 | 18.22 | 2.08 |
| AdvPT Zhang et al. (2024) | 23.50 | 0.33 | 63.70 | 3.47 | 19.47 | 3.30 | 43.10 | 0.87 | 12.23 | 0.00 | 28.57 | 0.60 | 26.33 | 0.40 | 25.77 | 0.27 | 30.33 | 1.16 |
| AdvVLP Zhou et al. (2024) | 53.23 | 22.10 | 87.33 | 62.97 | 33.43 | 18.60 | 78.80 | 40.83 | 55.80 | 17.83 | 49.77 | 25.23 | 52.80 | 21.67 | 51.50 | 22.10 | 57.83 | 28.92 |
| AdvMaPLe Khattak et al. (2023) | 52.93 | 21.90 | 88.23 | 64.90 | 30.87 | 17.50 | 77.87 | 42.83 | 56.67 | 18.53 | 52.90 | 28.73 | 52.53 | 21.90 | 50.97 | 23.20 | 57.87 | 29.94 |
| FAP Zhou et al. (2024) | 52.53 | 22.90 | 87.80 | 65.43 | 30.93 | 16.93 | 78.20 | 43.77 | 55.83 | 19.60 | 51.20 | 27.23 | 52.47 | 22.40 | 51.73 | 23.77 | 57.59 | 30.25 |
| AdvCLIP-LoRA (Ours) | 66.90 | 26.51 | 89.57 | 69.05 | 34.40 | 21.63 | 82.34 | 42.11 | 73.27 | 17.39 | 48.80 | 24.12 | 58.01 | 27.84 | 58.50 | 20.57 | 63.97 | 31.15 |
| *Relative Improvement* | *+25.68* | *+15.76* | *+1.52* | *+5.53* | *+2.9* | *+16.29* | *+4.49* | *−3.79* | *+29.29* | *−11.28* | *−7.75* | *−16.05* | *+9.87* | *+24.29* | *+13.09* | *−13.46* | *+10.54* | *+2.98* |

accuracy (5.56% below zero-shot CLIP) while achieving state-of-the-art robustness, yielding the best overall trade-off.

**Comparison with the Non-Robust Counterpart.** We compare AdvCLIP-LoRA with its non-robust variant, CLIP-LoRA, using the ViT-B/16 backbone in the 16-shot setting. As shown in Fig. 4 (top-left), for moderately small values of $\tau$, AdvCLIP-LoRA attains clean accuracy only marginally below CLIP-LoRA while achieving substantial gains in PGD accuracy, yielding a favorable robustness–accuracy trade-off. In practice, careful tuning of $\tau$ yields strong robustness gains at minimal nominal performance cost; we analyze this trade-off in more depth later. We provide an extensive comparison of CLIP-LoRA and AdvCLIP-LoRA on ViT-B/16 and ViT-B/32 across different shot counts in Appendix B.2.

## 5.3 Ablation Study

**LoRA Rank.** Fig. 4 (bottom-left) plots clean and PGD-100 accuracy on ImageNet-1K as a function of the LoRA rank $r$ for $\{1, 2, 4, 8, 16\}$ shots. Increasing the rank to a moderate value (e.g., $r = 16$) improves both clean and robust performance of AdvCLIP-LoRA across all shot counts, with the gains most pronounced in the 1-shot regime where data are scarce. To keep the number of trainable parameters low, we adopt $r = 2$ in the main experiments; despite its small footprint, this setting provides strong performance and a favorable robustness–accuracy trade-off, outperforming prompt tuning baselines.

**Attack Budget $\epsilon$.** Fig. 4 (top-right) shows the effect of the PGD budget $\epsilon$ on the average robust accuracy over five datasets using ViT-B/16. As expected, larger $\epsilon$ degrades robustness. Increasing the number of inner maximization steps $\tau$ consistently improves performance across budgets, yielding higher PGD accuracy for different $\epsilon$. Per-dataset and per-shot curves are provided in Fig. 6 (Appendix).

**Number of Inner Maximization Iterations $\tau$.** Figure 4 (bottom-right) shows clean and PGD-100 accuracy, averaged over eight datasets, as a function of the inner maximization steps $\tau$ in Alg. 1. Increasing $\tau$ tightens the approximation to the inner maximization in the minimax objective (Eq. 5), yielding steadily higher robustness; the largest gains occur between $\tau = 2$ and $\tau = 15$. This improvement comes at the cost of longer training and a modest drop in clean accuracy. For a fair comparison with baselines, we use $\tau = 2$ in the main tables; however, the curves indicate that $\tau \approx 15$ offers a strong robustness–efficiency trade-off, while for larger $\tau$ (beyond $\sim 15$) changes in both clean

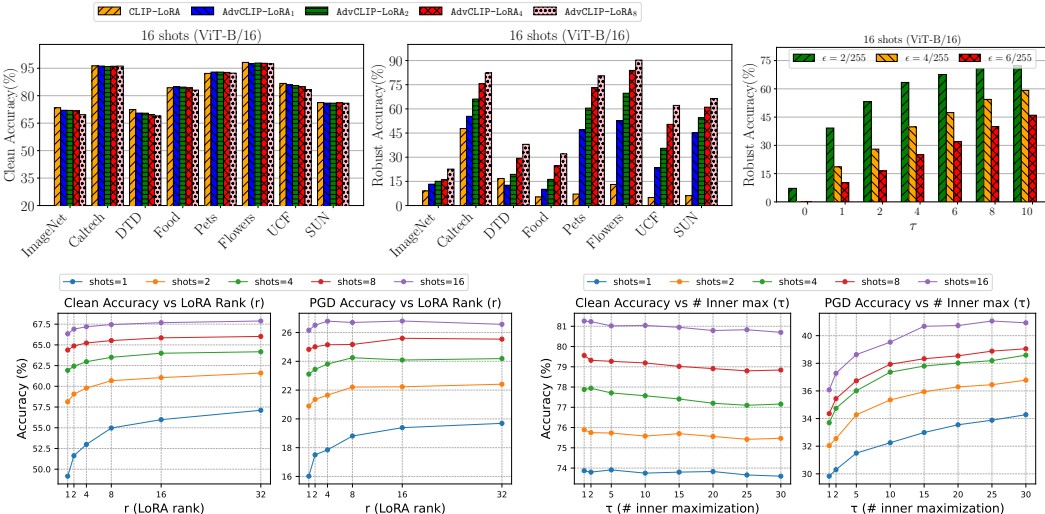

Figure 4: **Top-left:** comparison to the non-robust CLIP-LoRA. **Ablations for AdvCLIP-LoRA. Top-right:** effect of the PGD budget $\epsilon$. **Bottom-left:** effect of LoRA rank $r$. **Bottom-right:** effect of inner maximization steps $\tau$.

and PGD accuracy become less pronounced, particularly for the larger shots. In practice, $\tau \in [10, 15]$ is a reasonable default, with smaller $\tau$ remaining competitive under tight compute budgets.

**Ablation on LoRA Design Choices.** We study how different adapter configurations affect robustness and accuracy. In our default setup, LoRA is applied to both vision and text encoders, across all layers, and to attention projections. We vary one factor at a time and report averages over four datasets (clean, PGD-100, and harmonic mean) in Table 4. We observe that **(1)** restricting adapters to the vision encoder degrades performance, indicating the benefit of adapting both modalities, **(2)** placing adapters only at specific depths (e.g., up, bottom, mid, or half-stacks) underperforms using adapters in all layers, suggesting that distributed adaptation is more effective, **(3)** among per-matrix targets, applying LoRA to

Table 4: Average Clean, PGD-100, and harmonic mean (HM) for LoRA variants.

| Method | Clean | PGD-100 | HM |
|---|---|---|---|
| AdvCLIP-LoRA | **81.25** | **34.76** | **48.69** |
| Vision | 78.71 | 30.74 | 44.21 |
| $W_q$ | 80.65 | 30.62 | 44.39 |
| $W_v$ | 80.95 | 34.73 | 48.61 |
| $W_q W_v$ | 80.95 | 34.65 | 48.53 |
| up | 81.21 | 29.32 | 43.08 |
| bottom | 80.09 | 33.02 | 46.76 |
| half-up | 81.37 | 30.72 | 44.60 |
| half-bottom | 79.80 | 32.70 | 46.39 |
| mid | 80.45 | 30.98 | 44.73 |

the value projections ($W_v$) is the strongest single choice and nearly matches the full AdvCLIP-LoRA, while $W_q$ alone is weaker. Overall, the full configuration yields the best harmonic mean, reinforcing the importance of multi-modal, all-layer adaptation with appropriately chosen attention targets.

## 6 CONCLUSION

We introduced AdvCLIP-LoRA, a parameter-efficient adversarial fine-tuning method for CLIP that optimizes a minimax objective over low-rank adapters and an adversarial perturbation. Across eight datasets and two backbones (ViT-B/16 and ViT-B/32), the method achieves state-of-the-art results in few-shot classification, adversarial base-to-new generalization, and cross-dataset transfer, consistently improving PGD robustness while largely preserving clean accuracy. In contrast to adversarial prompt-tuning baselines, AdvCLIP-LoRA avoids large losses in clean accuracy and delivers strong robustness from the start. Ablations on adapter placement, LoRA rank, the attack budget $\epsilon$, and the number of inner maximization steps $\tau$ provide pragmatic guidance: adapting both encoders across all layers is beneficial, rank as small as $r = 2$ remains competitive, and $\tau$ around 15 offers a favorable robustness–efficiency trade-off. Finally, under standard assumptions, we establish convergence of the primal objective to a stationary point, giving a theoretical foundation for the proposed training procedure.

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

# A    RELATED WORK

## A.1    PARAMETER-EFFICIENT FINE-TUNING ON VLMS

Vision-Language Models (VLMs) such as LLaVa Liu et al. (2024) and GPT-4V Achiam et al. (2023) have achieved remarkable performance across various vision-language tasks, including cross-modal retrieval Hao & Zhang (2023); Hao et al. (2023) and image captioning Li et al. (2022a). However, these models typically contain billions of trainable parameters, making full fine-tuning (FFT) computationally expensive and inefficient, particularly for task-specific adaptations. To address this, Parameter-Efficient Fine-Tuning (PEFT) methods have been introduced, enabling adaptation with significantly fewer trainable parameters while maintaining performance close to FFT. PEFT techniques can be broadly categorized into adapter-based Houlsby et al. (2019); He et al., prompt-based Lester et al. (2021); Zhou et al. (2022), and Low-Rank Adaptation (LoRA)-based Hu et al. (2021); Zhao et al. approaches. Among these, LoRA stands out for its efficiency, effectiveness, and adaptability, making it a compelling choice for fine-tuning VLMs. In this work, we specifically focus on improving the robustness of LoRA against adversarial attacks.

## A.2    ROBUST FINE-TUNING

Robust fine-tuning (RFT) has been introduced as an efficient and cost-effective method for enhancing adversarial robustness in downstream tasks by adapting pre-trained feature extractors (FEs) through adversarial training data Shafahi et al.; Madry et al. (2018). The vanilla RFT jointly learns representations from both natural and adversarial data Shafahi et al.. This approach has been widely employed in fine-tuning adversarially self-supervised pre-trained models, demonstrating significant robustness improvements across various tasks Yu et al. (2022); Xu et al. (2023b). Expanding on this, TWINS Liu et al. (2023b) introduces a dual-network fine-tuning framework that enhances both generalization and robustness by optimizing two neural networks. More recently, AutoLoRA Xu et al. (2023a) refines RFT by decoupling the optimization process into two distinct components: using the LoRA branch for natural objectives while leveraging the FEs for adversarial objectives, thereby addressing the gradient instability present in TWINS. However, despite their effectiveness, these methods demand substantial computational resources due to intensive gradient computations and full model fine-tuning, making them impractical for VLMs.

## A.3    ADVERSARIAL ADAPTATION ON VLMS

It has been shown that VLMs are susceptible to adversarial attacks, where small input perturbations can cause them to make incorrect predictions with high confidence Zhao et al. (2023). Early approaches, such as Gan et al. (2020), employed adversarial training techniques to train VLMs from scratch, while others, like Yuan et al. (2024), sought to enhance adversarial robustness in downstream tasks by fine-tuning model parameters focusing only on visual models. More recently, studies Li et al. (2024); Zhang et al. (2024); Jia et al. (2025) have explored prompt tuning as a means of adversarial adaptation. For instance, APT Li et al. (2024) improves VLM robustness by learning robust textual prompts rather than modifying model weights. However, LoRA-based methods for strengthening VLM robustness in few-shot settings remain largely unexplored. Prior work in this area Ji et al. (2024) applies LoRA to adversarial fine-tuning with BLIP Li et al. (2022a), and does not provide theoretical guarantees. Our study differs in three key aspects: (i) we target few-shot learning with CLIP, (ii) we offer comprehensive comparisons against strong prompt-tuning baselines across multiple evaluation settings, and (iii) we conduct an extensive ablation study. In addition, we adopt a principled minimax optimization framework to enhance robustness and furnish a rigorous convergence analysis to a stationary solution.

# B  ADDITIONAL EXPERIMENTS RESULTS

## B.1  ADVERSARIAL FEW-SHOT LEARNING

Table 5: Detailed comparative analysis of various adversarial PEFT methods with ViT-B/32 as backbone. Top-1 accuracy averaged over 3 random seeds is reported.

| Shots | Method | Average | | ImageNet | | Caltech | | DTD | | Food | | Pets | | Flowers | | UCF | | SUN | |
|---|---|---|---|---|---|---|---|---|---|---|---|---|---|---|---|---|---|---|---|
| | | Clean | PGD | Clean | PGD | Clean | PGD | Clean | PGD | Clean | PGD | Clean | PGD | Clean | PGD | Clean | PGD | Clean | PGD |
| 16 | AdvVP Mao et al. | 41.90 | 17.13 | 46.27 | 12.77 | 90.40 | 52.60 | 29.20 | 13.87 | 1.07 | 0.80 | 56.40 | 16.43 | 56.17 | 22.03 | 0.97 | 0.93 | 54.70 | 17.63 |
| | APT Li et al. (2024) | 71.05 | 8.35 | 52.63 | 2.07 | 92.93 | 30.23 | 54.93 | 10.47 | 62.50 | 2.63 | 83.70 | 4.40 | 86.63 | 8.97 | 69.40 | 4.40 | 65.67 | 3.67 |
| | AdvPT Zhang et al. (2024) | 40.94 | 2.68 | 24.53 | 1.47 | 68.70 | 9.63 | 43.77 | 5.70 | 18.47 | 0.73 | 46.27 | 0.23 | 56.03 | 0.80 | 36.60 | 0.53 | 33.13 | 2.37 |
| | AdvMaPLe Khattak et al. (2023) | 71.48 | 38.11 | 52.93 | 21.90 | 92.17 | 68.63 | 57.93 | 32.73 | 65.13 | 25.27 | 83.27 | 36.87 | 87.87 | 58.70 | 68.97 | 31.67 | 63.57 | 29.70 |
| | AdvVLP Zhou et al. (2024) | 68.76 | 37.01 | 53.23 | 22.10 | 92.37 | 67.97 | 57.53 | 32.73 | 43.30 | 16.50 | 82.93 | 35.57 | 87.70 | 58.70 | 69.10 | 32.80 | 63.90 | 29.70 |
| | FAP Zhou et al. (2024) | 69.88 | 39.22 | 52.53 | 22.90 | 91.10 | 67.33 | 55.17 | 31.33 | 64.03 | 26.67 | 81.90 | 41.00 | 86.27 | 61.47 | 65.70 | 32.80 | 62.37 | 30.27 |
| | AdvCLIP-LoRA ($\tau = 1$) | 81.26 | 36.08 | 68.42 | 25.05 | 95.29 | 72.66 | 67.49 | 26.95 | 77.88 | 16.83 | 88.25 | 32.52 | 96.67 | 52.46 | 81.89 | 30.00 | 74.17 | 32.20 |
| | AdvCLIP-LoRA ($\tau = 2$) | 81.23 | 37.28 | 68.38 | 25.86 | 94.93 | 72.98 | 67.67 | 28.37 | 77.81 | 17.76 | 88.44 | 34.29 | 96.47 | 54.69 | 81.87 | 30.74 | 74.23 | 33.52 |
| | AdvCLIP-LoRA ($\tau = 5$) | 81.02 | 38.63 | 68.28 | 27.05 | 95.05 | 75.42 | 67.20 | 28.31 | 77.65 | 18.98 | 88.01 | 34.59 | 96.39 | 55.62 | 81.34 | 34.10 | 74.22 | 34.98 |
| | AdvCLIP-LoRA ($\tau = 10$) | 81.04 | 39.53 | 68.12 | 28.06 | 94.97 | 76.63 | 67.20 | 29.96 | 77.32 | 19.55 | 88.20 | 36.33 | 96.27 | 55.83 | 82.18 | 34.10 | 74.11 | 35.76 |
| | AdvCLIP-LoRA ($\tau = 15$) | 80.95 | 40.68 | 67.97 | 28.59 | 94.93 | 76.96 | 67.32 | 31.97 | 77.38 | 20.40 | 87.68 | 37.37 | 96.31 | 57.49 | 82.10 | 36.16 | 73.91 | 36.51 |
| | AdvCLIP-LoRA ($\tau = 20$) | 80.79 | 40.73 | 67.90 | 28.85 | 94.97 | 77.00 | 66.84 | 31.74 | 77.22 | 20.37 | 87.87 | 37.61 | 96.22 | 57.21 | 81.26 | 36.08 | 74.03 | 37.01 |
| | AdvCLIP-LoRA ($\tau = 25$) | 80.83 | 41.06 | 67.88 | 29.03 | 94.93 | 76.96 | 67.32 | 32.39 | 76.97 | 20.82 | 87.44 | 37.78 | 96.35 | 57.33 | 81.68 | 36.64 | 74.09 | 37.55 |
| | AdvCLIP-LoRA ($\tau = 30$) | 80.70 | 40.93 | 67.82 | 29.25 | 95.21 | 77.04 | 66.73 | 30.73 | 76.86 | 20.72 | 87.54 | 38.05 | 96.31 | 58.18 | 81.15 | 36.08 | 73.98 | 37.42 |
| | *Relative Improvement* | +13.37 | +0.79 | +27.97 | +22.53 | +2.2 | +11.66 | +16 | −8.46 | +18.72 | −26.7 | +5.38 | −11.39 | +9.5 | −9.18 | +18.41 | +3.96 | +12.85 | +18.14 |
| 8 | AdvVP Mao et al. | 43.24 | 17.21 | 46.37 | 11.90 | 91.20 | 50.33 | 23.63 | 11.47 | 1.00 | 0.83 | 57.43 | 17.33 | 55.50 | 23.57 | 18.27 | 4.93 | 52.53 | 17.30 |
| | APT Li et al. (2024) | 69.76 | 7.56 | 52.03 | 1.80 | 92.37 | 30.83 | 54.43 | 8.70 | 61.57 | 2.33 | 82.87 | 3.10 | 84.00 | 6.00 | 66.53 | 4.30 | 64.30 | 3.40 |
| | AdvPT Zhang et al. (2024) | 38.50 | 2.39 | 24.30 | 0.87 | 68.07 | 10.10 | 37.47 | 4.20 | 16.97 | 0.10 | 44.20 | 0.27 | 51.13 | 0.87 | 33.43 | 1.40 | 32.47 | 1.33 |
| | AdvMaPLe Khattak et al. (2023) | 62.90 | 30.45 | 52.13 | 20.60 | 90.63 | 63.80 | 33.20 | 16.97 | 62.70 | 20.13 | 55.60 | 21.07 | 83.10 | 48.80 | 64.33 | 25.93 | 61.50 | 26.30 |
| | AdvVLP Zhou et al. (2024) | 68.32 | 32.87 | 52.17 | 21.53 | 89.63 | 62.50 | 51.83 | 25.77 | 61.73 | 19.33 | 80.67 | 29.63 | 83.90 | 50.90 | 64.07 | 26.97 | 61.33 | 26.23 |
| | FAP Zhou et al. (2024) | 67.23 | 34.26 | 52.17 | 21.53 | 89.63 | 62.50 | 52.13 | 25.77 | 61.80 | 23.20 | 79.47 | 34.57 | 81.53 | 52.63 | 60.70 | 26.67 | 60.40 | 27.23 |
| | AdvCLIP-LoRA ($\tau = 1$) | 79.56 | 34.36 | 67.24 | 23.65 | 94.28 | 70.75 | 64.72 | 25.12 | 77.17 | 15.79 | 87.95 | 32.57 | 92.73 | 48.52 | 80.12 | 27.70 | 72.26 | 30.81 |
| | AdvCLIP-LoRA ($\tau = 2$) | 79.32 | 35.44 | 67.11 | 24.49 | 94.60 | 72.09 | 63.24 | 26.42 | 77.03 | 16.93 | 87.71 | 33.63 | 92.49 | 48.68 | 80.31 | 29.16 | 72.09 | 32.08 |
| | AdvCLIP-LoRA ($\tau = 5$) | 79.27 | 36.73 | 67.16 | 25.68 | 94.56 | 72.90 | 63.36 | 27.90 | 76.77 | 18.76 | 87.54 | 34.83 | 92.61 | 49.98 | 80.20 | 30.69 | 71.97 | 33.09 |
| | AdvCLIP-LoRA ($\tau = 10$) | 79.19 | 37.93 | 67.00 | 26.61 | 94.20 | 74.24 | 63.06 | 28.90 | 76.32 | 19.87 | 87.76 | 35.35 | 92.57 | 51.48 | 80.41 | 32.59 | 72.17 | 34.38 |
| | AdvCLIP-LoRA ($\tau = 15$) | 79.02 | 38.33 | 66.74 | 27.08 | 94.28 | 74.48 | 63.00 | 29.43 | 76.13 | 20.33 | 87.41 | 36.09 | 92.53 | 51.81 | 80.23 | 32.78 | 71.87 | 34.62 |
| | AdvCLIP-LoRA ($\tau = 20$) | 78.91 | 38.54 | 66.67 | 27.36 | 93.87 | 75.38 | 63.12 | 29.67 | 75.78 | 20.57 | 87.08 | 35.38 | 92.61 | 52.42 | 80.02 | 32.33 | 72.11 | 35.22 |
| | AdvCLIP-LoRA ($\tau = 25$) | 78.80 | 38.88 | 66.64 | 27.61 | 94.00 | 75.09 | 62.59 | 30.56 | 75.91 | 21.31 | 87.00 | 35.40 | 92.33 | 52.09 | 79.83 | 33.31 | 72.11 | 35.68 |
| | AdvCLIP-LoRA ($\tau = 30$) | 78.84 | 39.05 | 66.61 | 27.82 | 93.91 | 75.33 | 62.94 | 30.44 | 75.71 | 21.33 | 87.27 | 36.33 | 92.12 | 52.42 | 80.07 | 33.10 | 72.11 | 35.64 |
| | *Relative Improvement* | +13.52 | +10.71 | +26.82 | +23.59 | +1.98 | +16.36 | +15.86 | +12.15 | +21.72 | −14.35 | +5.9 | +2.26 | +10.2 | −2.19 | +20.86 | +20.84 | +12.24 | +26.26 |
| 4 | AdvVP Mao et al. | 43.10 | 16.40 | 49.80 | 11.13 | 90.17 | 52.50 | 22.73 | 9.27 | | | 57.80 | 16.20 | 55.97 | 23.73 | 1.07 | 0.80 | 48.47 | 13.03 |
| | APT Li et al. (2024) | 66.37 | 6.04 | 50.90 | 1.40 | 90.77 | 26.67 | 51.33 | 6.33 | 54.80 | 1.63 | 71.83 | 2.10 | 82.40 | 4.23 | 66.53 | 3.03 | 62.37 | 2.90 |
| | AdvPT Zhang et al. (2024) | 35.32 | 2.07 | 23.40 | 1.33 | 64.97 | 7.30 | 37.47 | 4.37 | 15.23 | 0.37 | 44.13 | 1.73 | 41.97 | 0.63 | 31.17 | 0.47 | 29.97 | 0.40 |
| | AdvMaPLe Khattak et al. (2023) | 51.01 | 21.61 | 51.27 | 19.00 | 89.53 | 59.40 | 6.43 | 2.40 | 60.00 | 14.83 | 30.70 | 9.03 | 52.20 | 25.37 | 59.73 | 21.30 | 58.23 | 21.53 |
| | AdvVLP Zhou et al. (2024) | 55.18 | 23.40 | 51.30 | 19.37 | 89.37 | 59.07 | 22.97 | 10.33 | 41.50 | 11.20 | 67.43 | 18.47 | 51.00 | 25.80 | 59.97 | 21.77 | 60.70 | 21.17 |
| | FAP Zhou et al. (2024) | 57.51 | 24.60 | 51.53 | 19.60 | 87.57 | 57.33 | 31.27 | 8.07 | 59.37 | 23.20 | 79.47 | 34.57 | 81.53 | 52.63 | 66.37 | 26.67 | 60.40 | 27.23 |
| | AdvCLIP-LoRA ($\tau = 1$) | 77.88 | 33.70 | 66.38 | 22.92 | 94.08 | 69.78 | 61.17 | 26.36 | 75.91 | 16.40 | 87.05 | 32.22 | 90.99 | 48.27 | 76.37 | 24.08 | 71.05 | 29.55 |
| | AdvCLIP-LoRA ($\tau = 2$) | 77.94 | 34.74 | 66.34 | 23.78 | 93.96 | 71.03 | 62.41 | 26.36 | 75.80 | 17.69 | 87.03 | 32.98 | 90.70 | 48.72 | 76.18 | 26.22 | 71.09 | 31.11 |
| | AdvCLIP-LoRA ($\tau = 5$) | 77.71 | 36.01 | 66.10 | 24.84 | 93.87 | 72.21 | 61.70 | 27.96 | 75.41 | 18.76 | 86.78 | 33.91 | 90.58 | 50.95 | 76.13 | 26.86 | 71.08 | 32.62 |
| | AdvCLIP-LoRA ($\tau = 10$) | 77.57 | 37.36 | 65.96 | 25.58 | 93.91 | 73.59 | 61.11 | 28.43 | 75.06 | 20.48 | 87.00 | 35.90 | 90.17 | 52.33 | 76.37 | 28.97 | 71.00 | 33.57 |
| | AdvCLIP-LoRA ($\tau = 15$) | 77.41 | 37.80 | 65.91 | 26.15 | 94.20 | 73.91 | 61.11 | 29.14 | 74.82 | 20.99 | 86.97 | 36.20 | 89.93 | 52.54 | 75.55 | 29.58 | 70.82 | 33.85 |
| | AdvCLIP-LoRA ($\tau = 20$) | 77.20 | 38.02 | 65.87 | 26.50 | 93.75 | 73.75 | 60.28 | 29.14 | 74.73 | 21.13 | 86.75 | 35.30 | 90.01 | 53.88 | 75.65 | 30.06 | 70.81 | 34.43 |
| | AdvCLIP-LoRA ($\tau = 25$) | 77.10 | 38.19 | 65.81 | 26.67 | 93.91 | 74.32 | 60.34 | 29.20 | 74.62 | 21.83 | 85.75 | 35.40 | 90.05 | 53.02 | 75.44 | 30.61 | 70.86 | 34.49 |
| | AdvCLIP-LoRA ($\tau = 30$) | 77.16 | 38.59 | 65.77 | 26.88 | 93.59 | 74.77 | 60.22 | 29.64 | 74.42 | 21.86 | 86.40 | 36.47 | 90.38 | 54.53 | 75.89 | 30.69 | 70.64 | 34.90 |
| | *Relative Improvement* | +16.88 | +51.87 | +28 | +30.51 | +3.46 | +23.89 | +19.05 | +163.84 | +26.43 | −11.72 | +9.48 | +3.85 | +9.43 | −0.57 | +14.79 | +8.62 | +13.84 | +23.28 |
| 2 | AdvVP Mao et al. | 39.24 | 16.23 | 46.23 | 10.90 | 91.25 | 55.23 | 14.27 | 6.93 | 1.05 | 0.10 | 47.13 | 15.10 | 61.47 | 26.93 | 1.73 | 1.07 | 59.20 | 13.57 |
| | APT Li et al. (2024) | 63.56 | 5.90 | 48.83 | 1.03 | 89.70 | 31.70 | 45.57 | 4.27 | 60.17 | 0.87 | 72.87 | 1.07 | 67.17 | 3.10 | 65.00 | 3.10 | 59.20 | 2.03 |
| | AdvPT Zhang et al. (2024) | 32.47 | 1.76 | 22.37 | 0.77 | 66.07 | 8.33 | 24.57 | 2.43 | 11.13 | 0.17 | 39.17 | 1.17 | 38.47 | 0.23 | 29.37 | 0.23 | 28.57 | 0.77 |
| | AdvMaPLe Khattak et al. (2023) | 39.09 | 15.58 | 49.97 | 17.13 | 88.00 | 56.20 | 16.53 | 4.20 | 3.10 | 0.67 | 34.03 | 6.87 | 46.17 | 17.00 | 21.17 | 6.20 | 53.73 | 16.33 |
| | AdvVLP Zhou et al. (2024) | 42.79 | 17.76 | 50.53 | 17.50 | 87.60 | 55.33 | 18.37 | 7.17 | 1.53 | 1.10 | 31.27 | 7.07 | 62.43 | 25.17 | 36.83 | 11.43 | 53.77 | 17.33 |
| | FAP Zhou et al. (2024) | 55.18 | 18.14 | 48.53 | 17.83 | 87.73 | 53.90 | 18.40 | 4.33 | 56.90 | 10.53 | 64.23 | 12.67 | 53.10 | 19.57 | 58.50 | 7.03 | 54.07 | 19.30 |
| | AdvCLIP-LoRA ($\tau = 1$) | 75.89 | 32.04 | 65.69 | 21.70 | 93.55 | 69.41 | 57.98 | 23.88 | 76.04 | 15.74 | 86.18 | 33.99 | 84.86 | 40.19 | 74.52 | 23.50 | 68.29 | 27.90 |
| | AdvCLIP-LoRA ($\tau = 2$) | 75.75 | 32.55 | 65.74 | 22.58 | 93.35 | 69.94 | 57.15 | 23.52 | 76.00 | 16.92 | 86.32 | 34.31 | 84.57 | 39.14 | 74.39 | 24.95 | 68.50 | 29.01 |
| | AdvCLIP-LoRA ($\tau = 5$) | 75.73 | 34.27 | 65.60 | 24.03 | 93.27 | 71.76 | 57.51 | 24.47 | 75.64 | 18.46 | 86.43 | 35.95 | 85.18 | 41.78 | 73.57 | 26.51 | 68.67 | 31.20 |
| | AdvCLIP-LoRA ($\tau = 10$) | 75.58 | 35.35 | 65.47 | 24.80 | 93.23 | 73.47 | 57.45 | 25.24 | 75.22 | 19.48 | 86.40 | 36.79 | 85.02 | 43.65 | 73.33 | 27.17 | 68.49 | 32.24 |
| | AdvCLIP-LoRA ($\tau = 15$) | 75.70 | 35.94 | 65.36 | 25.37 | 93.39 | 73.51 | 58.22 | 26.30 | 75.07 | 20.26 | 86.02 | 37.20 | 85.51 | 44.09 | 73.51 | 27.91 | 68.56 | 32.90 |
| | AdvCLIP-LoRA ($\tau = 20$) | 75.56 | 36.29 | 65.30 | 25.59 | 93.51 | 73.96 | 57.39 | 26.36 | 74.91 | 20.56 | 86.43 | 37.72 | 85.10 | 44.86 | 73.28 | 28.05 | 68.59 | 33.24 |
| | AdvCLIP-LoRA ($\tau = 25$) | 75.42 | 36.45 | 65.22 | 25.78 | 93.35 | 74.16 | 57.09 | 26.48 | 74.59 | 21.05 | 86.18 | 37.64 | 85.30 | 44.58 | 73.09 | 28.44 | 68.52 | 33.50 |
| | AdvCLIP-LoRA ($\tau = 30$) | 75.47 | 36.78 | 65.18 | 26.00 | 93.47 | 74.28 | 57.51 | 27.01 | 74.70 | 21.11 | 86.26 | 38.62 | 84.90 | 44.92 | 73.12 | 28.97 | 68.62 | 33.63 |
| | *Relative Improvement* | +18.91 | +94.87 | +29.57 | +39.09 | +2.17 | +30.73 | +26.07 | +163.84 | +25.01 | +85 | +18.57 | +143.64 | +26.57 | +62.09 | +12.82 | +137.71 | +15.69 | +67.05 |
| 1 | AdvVP Mao et al. | 43.62 | 17.94 | 46.60 | 11.07 | 85.73 | 50.33 | 24.43 | 5.23 | | | 57.60 | 22.73 | 63.10 | 29.70 | 3.37 | 0.40 | 41.20 | 11.10 |
| | APT Li et al. (2024) | 59.07 | 5.08 | 49.30 | 1.30 | 84.77 | 26.90 | 41.67 | 3.83 | 56.57 | 0.83 | 70.23 | 0.60 | 61.97 | 2.10 | 54.50 | 3.87 | 53.53 | 1.23 |
| | AdvPT Zhang et al. (2024) | 29.96 | 1.44 | 20.17 | 0.43 | 62.97 | 7.60 | 16.73 | 2.60 | 13.27 | 0.00 | 37.93 | 0.13 | 33.97 | 0.43 | 27.03 | 0.00 | 27.57 | 0.37 |
| | AdvMaPLe Khattak et al. (2023) | 33.52 | 11.38 | 49.27 | 14.60 | 85.53 | 48.37 | 13.63 | 2.93 | 5.27 | 0.30 | 30.67 | 4.97 | 1.40 | 0.10 | 32.70 | 7.07 | 49.70 | 12.67 |
| | AdvVLP Zhou et al. (2024) | 33.01 | 12.03 | 50.53 | 17.50 | 85.43 | 48.47 | 17.77 | 4.77 | 1.07 | 0.70 | 29.63 | 3.83 | 19.77 | 6.57 | 11.83 | 1.73 | 49.83 | 12.60 |
| | FAP Zhou et al. (2024) | 40.14 | 10.22 | 49.90 | 15.40 | 83.53 | 41.13 | 18.40 | 2.40 | 31.67 | 1.43 | 49.23 | 3.47 | 10.40 | 0.53 | 28.50 | 2.43 | 49.53 | 14.93 |
| | AdvCLIP-LoRA ($\tau = 1$) | 73.87 | 29.83 | 65.17 | 20.88 | 93.10 | 66.00 | 50.24 | 18.38 | 78.91 | 15.72 | 87.03 | 35.57 | 76.25 | 34.35 | 72.27 | 21.73 | 68.00 | 26.45 |
| | AdvCLIP-LoRA ($\tau = 2$) | 73.80 | 30.30 | 65.28 | 20.89 | 93.06 | 66.49 | 49.65 | 18.68 | 78.90 | 16.41 | 86.97 | 36.03 | 76.17 | 34.55 | 72.48 | 22.34 | 67.92 | 26.99 |
| | AdvCLIP-LoRA ($\tau = 5$) | 73.91 | 31.50 | 65.13 | 22.20 | 93.27 | 67.30 | 50.12 | 19.21 | 78.89 | 17.39 | 87.14 | 36.93 | 76.09 | 36.87 | 72.59 | 23.47 | 68.06 | 28.65 |
| | AdvCLIP-LoRA ($\tau = 10$) | 73.75 | 32.25 | 64.96 | 23.02 | 93.51 | 68.19 | 48.82 | 19.03 | 78.81 | 18.41 | 87.14 | 37.59 | 76.21 | 37.64 | 72.40 | 24.45 | 68.13 | 29.68 |
| | AdvCLIP-LoRA ($\tau = 15$) | 73.80 | 32.99 | 64.88 | 23.69 | 93.47 | 69.17 | 49.53 | 19.44 | 78.73 | 19.07 | 87.33 | 38.46 | 76.08 | 38.37 | 72.40 | 25.17 | 68.03 | 30.57 |
| | AdvCLIP-LoRA ($\tau = 20$) | 73.83 | 33.55 | 64.68 | 24.04 | 93.39 | 69.61 | 50.12 | 19.98 | 78.72 | 19.64 | 87.00 | 39.33 | 76.37 | 38.69 | 72.24 | 25.75 | 68.11 | 31.38 |
| | AdvCLIP-LoRA ($\tau = 25$) | 73.66 | 33.88 | 64.65 | 24.29 | 93.23 | 69.98 | 49.88 | 20.04 | 78.65 | 20.09 | 87.14 | 39.30 | 75.64 | 39.18 | 71.82 | 26.38 | 68.25 | 31.80 |
| | AdvCLIP-LoRA ($\tau = 30$) | 73.60 | 34.28 | 64.60 | 24.52 | 93.31 | 70.34 | 49.76 | 20.80 | 78.58 | 20.26 | 86.84 | 40.15 | 75.68 | 39.38 | 71.95 | 26.78 | 68.10 | 32.01 |
| | *Relative Improvement* | +24.85 | +79.77 | +28.56 | +31.54 | +9.07 | +35.49 | +17.16 | +47.18 | +39.31 | +163.84 | +24.08 | +65.38 | +20.78 | +26.73 | +32.84 | +163.84 | +27.27 | +98.79 |

This section expands Section 5.2 with full adversarial few-shot results on ViT-B/32 across $\{1, 2, 4, 8, 16\}$ shots. In addition to baseline comparisons, we sweep the number of inner maximization steps $\tau$ used to train AdvCLIP-LoRA. Baselines report training with a 2-step PGD procedure, and the impact of using more steps is unspecified; therefore, for fairness, the main paper fixes $\tau = 2$. The extended tables here show that increasing $\tau$ consistently improves AdvCLIP-LoRA's robustness and overall accuracy. Here, we also report $\Delta$, defined as the relative improvement of AdvCLIP-LoRA with $\tau = 10$ over the strongest non-ours baseline for each dataset and shot.

## B.2 COMPARATIVE ANALYSIS OF ADVCLIP-LORA AND CLIP-LORA

Table 6: Detailed results for the 8 datasets with ViT-B/16 as backbone. Top-1 accuracy averaged over 3 random seeds is reported. Highest value is highlighted in **bold**.

| Shots | Method | ImageNet Clean | FGSM | PGD | Caltech Clean | FGSM | PGD | DTD Clean | FGSM | PGD | Food Clean | FGSM | PGD |
|---|---|---|---|---|---|---|---|---|---|---|---|---|---|
| | CLIP-LoRA | **70.24** | 15.14 | 4.73 | **94.20** | 59.86 | 26.26 | **54.77** | 14.99 | 3.11 | **84.99** | 8.43 | 2.90 |
| | AdvCLIP-LoRA ($\tau=1$) | 56.02 | 29.17 | 17.10 | 92.67 | 62.70 | 26.40 | 49.64 | 20.09 | 4.06 | 79.86 | 26.50 | 9.62 |
| | AdvCLIP-LoRA ($\tau=2$) | 54.76 | 30.52 | 19.44 | 90.20 | 67.29 | 29.48 | 50.53 | 21.12 | 3.04 | 78.19 | 31.31 | 12.74 |
| 1 | AdvCLIP-LoRA ($\tau=4$) | 53.14 | **31.19** | 21.70 | 87.17 | 70.18 | 34.16 | 48.84 | 21.16 | 2.60 | 74.88 | 35.01 | 20.04 |
| | AdvCLIP-LoRA ($\tau=6$) | 50.19 | 30.96 | 21.30 | 83.96 | **79.69** | 37.09 | 44.71 | 31.86 | 3.17 | 72.09 | 57.40 | 26.45 |
| | AdvCLIP-LoRA ($\tau=8$) | 45.35 | 30.60 | 21.66 | 81.39 | 78.96 | **41.28** | 42.61 | 32.76 | 4.24 | 68.57 | **58.32** | 32.84 |
| | AdvCLIP-LoRA ($\tau=10$) | 42.88 | 30.12 | **22.38** | 77.51 | 76.54 | 40.76 | 42.12 | **33.35** | **6.14** | 64.52 | 56.22 | **34.47** |
| | CLIP-LoRA | **71.52** | 14.59 | 5.12 | 95.16 | 59.39 | 29.19 | **63.73** | 19.39 | 6.68 | 83.07 | 7.83 | 2.21 |
| | AdvCLIP-LoRA ($\tau=1$) | 67.81 | 40.62 | 37.74 | **95.28** | 76.84 | 61.49 | 59.73 | 27.64 | 8.89 | 83.75 | 31.57 | 27.47 |
| | AdvCLIP-LoRA ($\tau=2$) | 67.63 | 42.53 | 38.42 | 95.15 | 80.68 | 72.81 | 59.26 | 31.01 | 13.59 | **83.77** | 35.19 | 35.03 |
| 4 | AdvCLIP-LoRA ($\tau=4$) | 67.43 | 42.50 | 41.40 | 95.20 | 84.00 | 82.80 | 60.40 | 36.41 | 26.04 | 83.67 | 43.52 | 50.08 |
| | AdvCLIP-LoRA ($\tau=6$) | 66.90 | 44.35 | 43.75 | 95.19 | 92.03 | 87.21 | 59.75 | 49.45 | 34.71 | 83.53 | 69.85 | 56.92 |
| | AdvCLIP-LoRA ($\tau=8$) | 66.67 | 44.47 | 43.92 | 95.03 | **92.67** | 88.27 | 59.42 | 50.87 | 39.54 | 83.12 | **73.09** | 62.16 |
| | AdvCLIP-LoRA ($\tau=10$) | 65.93 | **45.15** | **45.07** | 95.03 | 92.66 | **89.36** | 59.60 | **52.42** | **44.48** | 82.56 | 72.74 | **65.41** |
| | CLIP-LoRA | **73.41** | 14.56 | 5.51 | **96.31** | 60.63 | 31.05 | **72.40** | 24.57 | 9.30 | 84.32 | 7.15 | 2.45 |
| | AdvCLIP-LoRA ($\tau=1$) | 72.03 | 44.41 | 30.24 | 96.19 | 79.92 | 74.13 | 70.51 | 33.06 | 15.78 | **84.77** | 26.43 | 23.41 |
| | AdvCLIP-LoRA ($\tau=2$) | 71.96 | 46.91 | 48.73 | 95.95 | 81.35 | 81.12 | 70.45 | 38.00 | 30.99 | 84.70 | 28.42 | 34.18 |
| 16 | AdvCLIP-LoRA ($\tau=4$) | 71.69 | 47.42 | 50.08 | 96.09 | 82.14 | 86.31 | 69.70 | 42.61 | 46.02 | 84.24 | 32.68 | 48.56 |
| | AdvCLIP-LoRA ($\tau=6$) | 71.32 | 47.44 | 50.34 | 96.08 | 93.12 | 88.95 | 69.31 | 60.26 | 52.27 | 83.68 | 66.18 | 55.57 |
| | AdvCLIP-LoRA ($\tau=8$) | 69.63 | 53.31 | 56.33 | 96.16 | 93.72 | 90.82 | 68.93 | 61.43 | 55.70 | 83.05 | 68.12 | 59.64 |
| | AdvCLIP-LoRA ($\tau=10$) | 67.00 | **54.71** | **57.56** | 96.09 | **94.28** | **91.98** | 68.28 | **62.61** | **58.69** | 82.75 | **69.25** | **62.17** |

| Shots | Method | Pets Clean | FGSM | PGD | Flowers Clean | FGSM | PGD | UCF Clean | FGSM | PGD | SUN Clean | FGSM | PGD |
|---|---|---|---|---|---|---|---|---|---|---|---|---|---|
| | CLIP-LoRA | **92.14** | 23.52 | 17.21 | **82.45** | 6.70 | 3.15 | **75.95** | 18.36 | 2.98 | **70.22** | 17.78 | 6.20 |
| | AdvCLIP-LoRA ($\tau=1$) | 90.02 | 23.51 | 17.17 | 70.62 | 26.04 | 5.33 | 66.44 | 29.53 | 8.94 | 61.68 | 35.60 | 17.50 |
| | AdvCLIP-LoRA ($\tau=2$) | 88.34 | 40.84 | 16.75 | 69.62 | 30.42 | 6.86 | 63.04 | 31.95 | 10.68 | 61.02 | 39.98 | 20.41 |
| 1 | AdvCLIP-LoRA ($\tau=4$) | 82.76 | 41.56 | 16.66 | 66.14 | 36.80 | 8.66 | 58.80 | 35.09 | 16.07 | 60.01 | 39.91 | 24.03 |
| | AdvCLIP-LoRA ($\tau=6$) | 78.35 | 40.96 | 17.90 | 62.79 | 39.09 | 8.86 | 54.59 | **37.02** | 18.71 | 58.61 | 41.34 | 27.40 |
| | AdvCLIP-LoRA ($\tau=8$) | 73.21 | **42.56** | 21.15 | 57.69 | **40.06** | **11.20** | 49.58 | 36.80 | **20.22** | 56.66 | 43.33 | 30.46 |
| | AdvCLIP-LoRA ($\tau=10$) | 66.37 | 40.95 | **22.92** | 54.01 | 39.29 | 10.79 | 45.33 | 34.65 | 19.57 | 54.56 | **43.80** | **31.46** |
| | CLIP-LoRA | 89.99 | 16.73 | 10.08 | **93.48** | 11.20 | 7.62 | **80.44** | 18.85 | 4.00 | **72.19** | 16.15 | 6.20 |
| | AdvCLIP-LoRA ($\tau=1$) | **91.36** | 57.37 | 51.38 | 91.10 | 46.41 | 31.14 | 74.42 | 37.49 | 25.23 | 70.99 | 45.40 | 40.31 |
| | AdvCLIP-LoRA ($\tau=2$) | 91.06 | 60.57 | 60.56 | 91.03 | 51.39 | 45.29 | 78.51 | 38.06 | 32.07 | 71.28 | 48.84 | 47.63 |
| 4 | AdvCLIP-LoRA ($\tau=4$) | 91.07 | 64.57 | 71.11 | 91.03 | 58.53 | 61.24 | 77.96 | 42.07 | 45.39 | 71.19 | 51.37 | 50.67 |
| | AdvCLIP-LoRA ($\tau=6$) | 91.05 | 67.77 | 77.72 | 90.62 | 65.16 | 69.60 | 77.83 | 45.35 | 52.36 | 71.69 | 56.71 | 56.20 |
| | AdvCLIP-LoRA ($\tau=8$) | 91.06 | 69.96 | 80.19 | 89.78 | 66.38 | 74.67 | 77.09 | 47.98 | 55.99 | 70.96 | 57.14 | 56.96 |
| | AdvCLIP-LoRA ($\tau=10$) | 91.22 | **71.70** | **82.02** | 89.35 | **68.59** | **77.75** | 76.60 | **50.47** | **58.53** | 71.04 | **60.27** | **59.89** |
| | CLIP-LoRA | 92.18 | 16.28 | 7.14 | **98.19** | 17.39 | 13.09 | **86.71** | 22.20 | 5.01 | **76.22** | 16.94 | 6.15 |
| | AdvCLIP-LoRA ($\tau=1$) | **92.90** | 48.31 | 46.94 | 97.55 | 57.42 | 52.53 | 85.96 | 37.73 | 23.54 | 75.94 | 48.77 | 45.10 |
| | AdvCLIP-LoRA ($\tau=2$) | 92.88 | 49.72 | 60.47 | 97.84 | 60.87 | 69.71 | 85.58 | 36.71 | 35.53 | 75.92 | 52.37 | 54.50 |
| 16 | AdvCLIP-LoRA ($\tau=4$) | 92.72 | 51.65 | 73.12 | 97.70 | 65.68 | 83.88 | 84.92 | 39.19 | 50.39 | 76.09 | 55.02 | 61.05 |
| | AdvCLIP-LoRA ($\tau=6$) | 92.65 | 56.37 | 78.18 | 97.45 | 68.71 | 88.09 | 84.33 | 40.60 | 58.42 | 75.58 | 57.18 | 64.04 |
| | AdvCLIP-LoRA ($\tau=8$) | 92.33 | 58.02 | 80.52 | 97.39 | 70.97 | 90.29 | 83.38 | 42.05 | 62.15 | 75.89 | 59.28 | 66.43 |
| | AdvCLIP-LoRA ($\tau=10$) | 92.43 | **60.49** | **81.86** | 97.33 | **74.26** | **91.83** | 83.08 | **43.93** | **65.40** | 75.87 | **61.92** | **68.18** |

**Setup.** For adversarial training, we define the projection set for updating $\delta$ as an $\ell_\infty$-ball with a radius of $\epsilon = 10/255$ across all datasets. To evaluate adversarial robustness, we implement two standard attack methods: FGSM Szegedy et al. (2013) and PGD Madry et al. (2018). For FGSM, we set $\epsilon = 10/255$, while for PGD, we use $\epsilon = 2/255$ with a total of 20 attack iterations.

Table 6 presents the experimental results of CLIP-LoRA and AdvCLIP-LoRA with varying values of $\tau$, using the ViT-B/16 backbone. Our findings show that AdvCLIP-LoRA significantly enhances model robustness, increasing FGSM accuracy for a minimum of 11.04% and a maximum of 42.97%, and PGD accuracy for a minimum of 15.67% and a maximum of 62.25%, averaged across all datasets. Specifically, for $\tau = 1$, the model demonstrates improved robustness without a significant impact on clean accuracy (the difference in clean accuracy is less than 22.58% for 1 shot and less than 2.24% for 16 shots, on average). As $\tau$ increases, robustness continues to improve; however, this comes at the cost of a slight decrease in clean accuracy. This effect is less prominent for larger shots. It is noteworthy that with 16 shots, the clean accuracy decreases by an average of only 2.24%, while we observe a minimum improvement of 24.55% in the FGSM robustness and 29.00% in the PGD robustness. For clearer comparison, we visualize clean and PGD-robust accuracies for both 4-shot and 16-shot settings across ViT-B/16 and ViT-B/32 backbones in Fig. 5. Further results using the ViT-B/32 model can be found in Table 7.

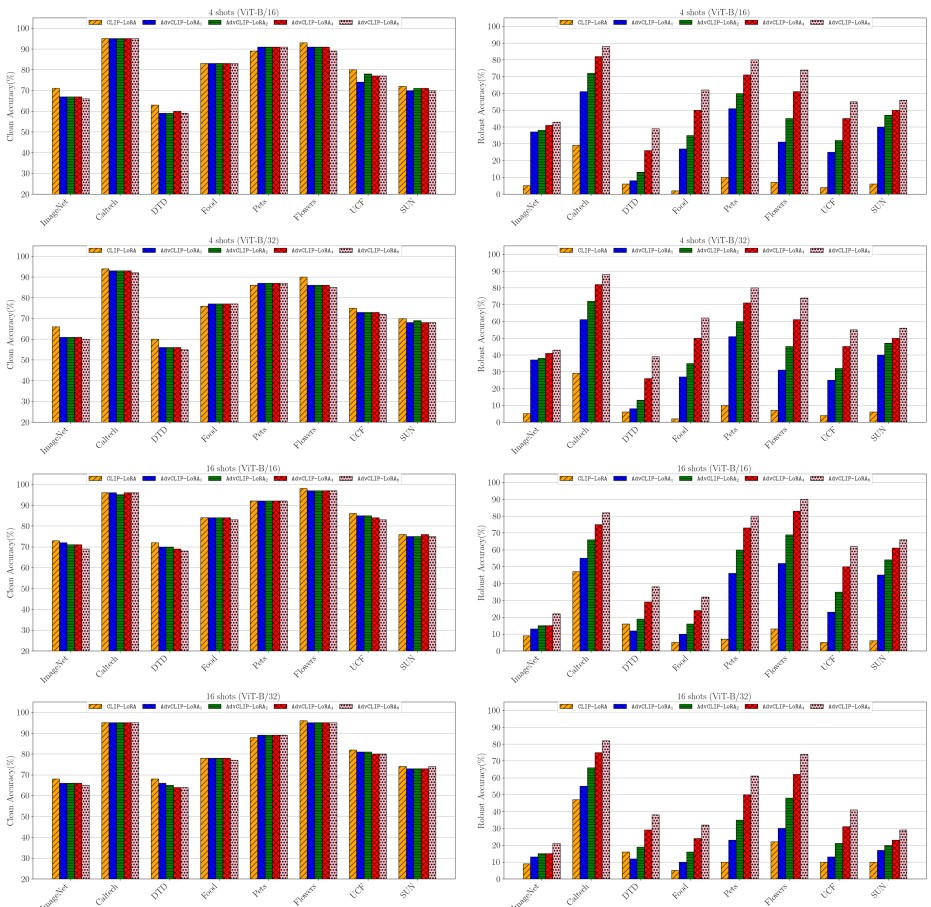

Figure 5: Comparative analysis of CLIP-LoRA and AdvCLIP-LoRA with ViT-B/16 and ViT-B/32 backbones on 8 fine-grained datasets, showing clean accuracy and PGD-adversarial robustness (shots labeled above). AdvCLIP-LoRA$_i$ means AdvCLIP-LoRA with $\tau = i$.

Table 7: Detailed results for the 8 datasets with ViT-B/32 as backbone. Top-1 accuracy averaged over 3 random seeds is reported. Highest value is highlighted in **bold**.

| | | ImageNet | | | Caltech | | | DTD | | | Food | | |
|---|---|---|---|---|---|---|---|---|---|---|---|---|---|
| Shots | Method | Clean | FGSM | PGD | Clean | FGSM | PGD | Clean | FGSM | PGD | Clean | FGSM | PGD |
| | CLIP-LoRA | **65.70** | 15.97 | 8.23 | **93.54** | 62.83 | 42.34 | **55.46** | 17.16 | **9.16** | **76.53** | 9.00 | 4.57 |
| | AdvCLIP-LoRA ($\tau=1$) | 56.97 | 21.00 | 11.88 | 92.11 | 64.44 | 40.04 | 52.03 | 17.83 | 5.28 | 75.68 | 14.17 | 6.83 |
| | AdvCLIP-LoRA ($\tau=2$) | 56.73 | 20.68 | 11.34 | 91.89 | 66.02 | 41.61 | 52.05 | 19.36 | 6.36 | 75.70 | 16.11 | 8.62 |
| 2 | AdvCLIP-LoRA ($\tau=4$) | 56.32 | 22.14 | 12.06 | 91.94 | 68.26 | 44.88 | 51.16 | 19.41 | 6.78 | 75.71 | 18.97 | 10.31 |
| | AdvCLIP-LoRA ($\tau=6$) | 55.45 | 23.21 | **12.48** | 91.63 | 70.45 | 46.69 | 50.26 | 20.75 | 7.25 | 76.11 | 21.26 | 11.93 |
| | AdvCLIP-LoRA ($\tau=8$) | 54.87 | **23.65** | 12.38 | 91.76 | 71.51 | 48.79 | 50.22 | 21.12 | 7.49 | 76.32 | 23.27 | 13.25 |
| | AdvCLIP-LoRA ($\tau=10$) | 53.46 | 22.27 | 10.85 | 91.58 | **74.28** | **52.32** | 49.33 | **21.49** | 8.18 | 76.35 | **25.05** | **14.85** |
| | CLIP-LoRA | **66.43** | 15.59 | 8.59 | **94.44** | 62.44 | 42.12 | **60.18** | 19.35 | 10.70 | 76.18 | 9.02 | 4.55 |
| | AdvCLIP-LoRA ($\tau=1$) | 61.60 | 20.63 | 13.03 | 93.90 | 64.46 | 43.28 | 56.40 | 18.99 | 7.53 | 77.30 | 14.00 | 7.96 |
| | AdvCLIP-LoRA ($\tau=2$) | 61.44 | 20.36 | 12.18 | 93.75 | 67.96 | 51.67 | 56.68 | 21.06 | 9.73 | 77.52 | 14.46 | 10.29 |
| 4 | AdvCLIP-LoRA ($\tau=4$) | 61.44 | 20.46 | 12.30 | 93.81 | 71.09 | 55.11 | 56.58 | 22.24 | 12.81 | 77.88 | 16.49 | 13.92 |
| | AdvCLIP-LoRA ($\tau=6$) | 60.49 | 20.80 | 12.77 | 93.47 | 85.94 | 59.67 | 56.17 | 36.90 | 15.62 | **77.96** | 49.43 | 17.54 |
| | AdvCLIP-LoRA ($\tau=8$) | 60.22 | 21.91 | 12.99 | 92.82 | 86.17 | 62.50 | 55.32 | 37.87 | 18.62 | 77.40 | 49.34 | 23.05 |
| | AdvCLIP-LoRA ($\tau=10$) | 59.10 | **22.65** | **13.57** | 92.94 | **86.49** | **65.52** | 54.34 | **38.67** | **22.02** | 76.91 | **50.40** | **27.20** |
| | CLIP-LoRA | **67.28** | 15.35 | 8.62 | 94.46 | 61.68 | 43.30 | **63.36** | 21.30 | 13.12 | 76.90 | 8.84 | 4.65 |
| | AdvCLIP-LoRA ($\tau=1$) | 64.19 | 22.24 | 14.53 | **94.67** | 65.44 | 49.37 | 61.17 | 20.57 | 9.99 | **78.03** | 12.35 | 8.47 |
| | AdvCLIP-LoRA ($\tau=2$) | 63.93 | 22.37 | 14.74 | 94.63 | 67.10 | 58.70 | 60.78 | 21.63 | 14.34 | 77.90 | 12.05 | 13.36 |
| 8 | AdvCLIP-LoRA ($\tau=4$) | 63.76 | 22.93 | 16.41 | 94.54 | 68.38 | 68.78 | 61.11 | 22.56 | 22.69 | 77.55 | 13.37 | 22.54 |
| | AdvCLIP-LoRA ($\tau=6$) | 63.50 | 24.00 | 17.57 | 94.28 | **69.90** | 74.21 | 60.05 | 23.15 | 27.88 | 77.29 | 14.98 | 27.55 |
| | AdvCLIP-LoRA ($\tau=8$) | 63.22 | **24.20** | 18.38 | 94.38 | 69.25 | 77.78 | 58.81 | 23.46 | 30.44 | 76.94 | 15.39 | 31.07 |
| | AdvCLIP-LoRA ($\tau=10$) | 62.74 | 23.69 | **18.51** | 94.39 | 68.45 | **79.68** | 58.91 | **23.62** | **32.29** | 76.57 | **16.25** | **33.24** |
| | CLIP-LoRA | **68.43** | 15.09 | 9.06 | 95.50 | 64.29 | 47.80 | **68.62** | 20.11 | 16.80 | 78.00 | 8.97 | 5.32 |
| | AdvCLIP-LoRA ($\tau=1$) | 66.24 | 19.48 | 13.26 | **95.84** | 67.46 | 55.38 | 66.90 | 22.40 | 12.61 | **78.55** | 12.96 | 10.10 |
| | AdvCLIP-LoRA ($\tau=2$) | 66.08 | 20.06 | 15.03 | 95.40 | 68.64 | 66.09 | 65.84 | 21.63 | 19.37 | 78.41 | 12.84 | 16.25 |
| 16 | AdvCLIP-LoRA ($\tau=4$) | 66.08 | 21.13 | 15.98 | 95.39 | 68.19 | 75.62 | 64.89 | 22.02 | 29.33 | 78.09 | 12.68 | 24.62 |
| | AdvCLIP-LoRA ($\tau=6$) | 65.39 | 22.46 | 17.10 | 95.46 | 88.52 | 80.22 | 63.91 | 43.04 | 34.02 | 77.75 | 45.41 | 28.79 |
| | AdvCLIP-LoRA ($\tau=8$) | 65.63 | 23.74 | **21.17** | 95.31 | 89.22 | 82.29 | 64.01 | 45.18 | 38.00 | 77.44 | 46.89 | 32.03 |
| | AdvCLIP-LoRA ($\tau=10$) | 64.06 | **24.07** | 17.93 | 95.28 | **89.59** | **84.10** | 64.77 | **46.69** | **39.26** | 77.08 | **48.62** | **35.18** |

| | | Pets | | | Flowers | | | UCF | | | SUN | | |
|---|---|---|---|---|---|---|---|---|---|---|---|---|---|
| Shots | Method | Clean | FGSM | PGD | Clean | FGSM | PGD | Clean | FGSM | PGD | Clean | FGSM | PGD |
| | CLIP-LoRA | **87.43** | 21.70 | 16.11 | **84.40** | 15.36 | 10.68 | **74.07** | 22.04 | 7.18 | **68.71** | 17.61 | 8.56 |
| | AdvCLIP-LoRA ($\tau=1$) | 85.70 | 34.83 | 16.92 | 77.71 | 19.48 | 8.10 | 69.41 | 26.69 | 8.48 | 65.45 | 23.28 | 13.56 |
| | AdvCLIP-LoRA ($\tau=2$) | 85.14 | 34.61 | 18.19 | 77.16 | 22.58 | 10.53 | 68.06 | 28.94 | 8.99 | 65.22 | 23.97 | 13.80 |
| 2 | AdvCLIP-LoRA ($\tau=4$) | 84.90 | 37.19 | 22.85 | 76.12 | 26.01 | 12.29 | 67.48 | 31.42 | 10.31 | 64.96 | 23.77 | 14.58 |
| | AdvCLIP-LoRA ($\tau=6$) | 84.67 | 40.80 | 26.93 | 75.78 | 28.49 | 13.52 | 66.56 | 33.71 | 11.86 | 64.64 | 25.18 | 14.62 |
| | AdvCLIP-LoRA ($\tau=8$) | 84.39 | 46.05 | 31.78 | 74.83 | 33.10 | 16.20 | 65.64 | 36.75 | 13.79 | 63.30 | 27.20 | 16.48 |
| | AdvCLIP-LoRA ($\tau=10$) | 85.07 | **49.10** | **34.16** | 72.71 | **37.89** | **19.16** | 64.19 | **40.73** | **16.70** | 63.59 | **29.12** | **17.01** |
| | CLIP-LoRA | 86.43 | 16.02 | 11.74 | **90.21** | 16.82 | 13.71 | **75.65** | 25.87 | 7.67 | **70.20** | 16.96 | 8.89 |
| | AdvCLIP-LoRA ($\tau=1$) | **87.87** | 34.51 | 27.58 | 86.32 | 20.46 | 16.83 | 73.43 | 25.87 | 10.09 | 68.93 | 24.03 | 15.60 |
| | AdvCLIP-LoRA ($\tau=2$) | 87.87 | 35.30 | 33.51 | 86.26 | 21.32 | 19.33 | 73.39 | 27.39 | 12.88 | 69.22 | 26.58 | 16.65 |
| 4 | AdvCLIP-LoRA ($\tau=4$) | 87.82 | 35.82 | 37.40 | 86.26 | 26.00 | 30.50 | 73.57 | 31.43 | 16.59 | 68.92 | 27.55 | 17.11 |
| | AdvCLIP-LoRA ($\tau=6$) | 87.80 | 37.40 | 46.76 | 86.29 | 30.50 | 32.46 | 73.72 | 33.87 | 23.55 | 68.88 | 30.48 | 19.27 |
| | AdvCLIP-LoRA ($\tau=8$) | 87.56 | 41.96 | 53.47 | 85.82 | 33.62 | 39.13 | 72.75 | 35.43 | 26.53 | 68.40 | 32.25 | 20.09 |
| | AdvCLIP-LoRA ($\tau=10$) | 87.52 | **43.52** | **56.88** | 85.34 | **37.54** | **43.78** | 72.28 | **37.15** | **29.27** | 68.47 | **38.04** | **23.22** |
| | CLIP-LoRA | 87.61 | 16.54 | 10.92 | **93.29** | 21.60 | 18.35 | **80.46** | 22.48 | 9.17 | **72.18** | 18.23 | 9.85 |
| | AdvCLIP-LoRA ($\tau=1$) | 88.71 | 30.46 | 24.04 | 91.76 | 28.11 | 21.26 | 78.64 | 26.55 | 11.77 | 71.73 | 24.53 | 16.43 |
| | AdvCLIP-LoRA ($\tau=2$) | **88.75** | 29.11 | 35.99 | 91.91 | 27.81 | 34.81 | 78.67 | 27.45 | 18.03 | 71.71 | 24.76 | 17.73 |
| 8 | AdvCLIP-LoRA ($\tau=4$) | 88.63 | 28.67 | 50.19 | 91.65 | 29.57 | 51.02 | 78.35 | **29.29** | 27.54 | 71.86 | 27.07 | 20.80 |
| | AdvCLIP-LoRA ($\tau=6$) | 88.65 | 30.79 | 57.28 | 91.76 | 33.65 | 58.67 | 77.53 | 28.86 | 33.02 | 71.57 | 29.72 | 23.87 |
| | AdvCLIP-LoRA ($\tau=8$) | 88.53 | 34.13 | 61.57 | 91.20 | 33.51 | 63.04 | 77.22 | 28.71 | 37.31 | 71.39 | **31.83** | 26.10 |
| | AdvCLIP-LoRA ($\tau=10$) | 88.26 | **35.15** | **64.59** | 90.91 | **35.49** | **65.77** | 76.36 | 28.15 | **39.32** | 71.10 | 31.77 | **28.14** |
| | CLIP-LoRA | 88.43 | 15.40 | 10.54 | **96.39** | 24.13 | 22.26 | **82.86** | 25.09 | 10.16 | **74.09** | 18.20 | 10.52 |
| | AdvCLIP-LoRA ($\tau=1$) | 89.67 | **27.06** | 23.70 | 95.22 | 32.45 | 30.33 | 81.18 | **27.36** | 13.95 | 73.77 | 24.73 | 17.79 |
| | AdvCLIP-LoRA ($\tau=2$) | 89.66 | 24.00 | 35.08 | 95.75 | 31.14 | 48.50 | 81.18 | 26.86 | 21.92 | 73.46 | 23.69 | 20.29 |
| 16 | AdvCLIP-LoRA ($\tau=4$) | **89.69** | 24.41 | 50.63 | 95.93 | 33.37 | 62.78 | 80.99 | 26.34 | 31.94 | 73.52 | 25.18 | 23.23 |
| | AdvCLIP-LoRA ($\tau=6$) | 89.56 | 24.81 | 57.38 | 95.49 | 34.89 | 70.13 | 80.49 | 25.48 | 37.94 | 73.61 | 27.10 | 25.11 |
| | AdvCLIP-LoRA ($\tau=8$) | 89.27 | 24.85 | 61.59 | 95.25 | 35.24 | 74.29 | 80.49 | 25.10 | 41.07 | 74.09 | 27.61 | 29.55 |
| | AdvCLIP-LoRA ($\tau=10$) | 88.83 | 25.10 | **64.06** | 95.20 | **36.64** | **77.37** | 79.56 | 25.85 | **43.64** | 73.65 | **31.34** | **31.08** |

### B.3 ABLATION ON ATTACK BUDGET $\epsilon$

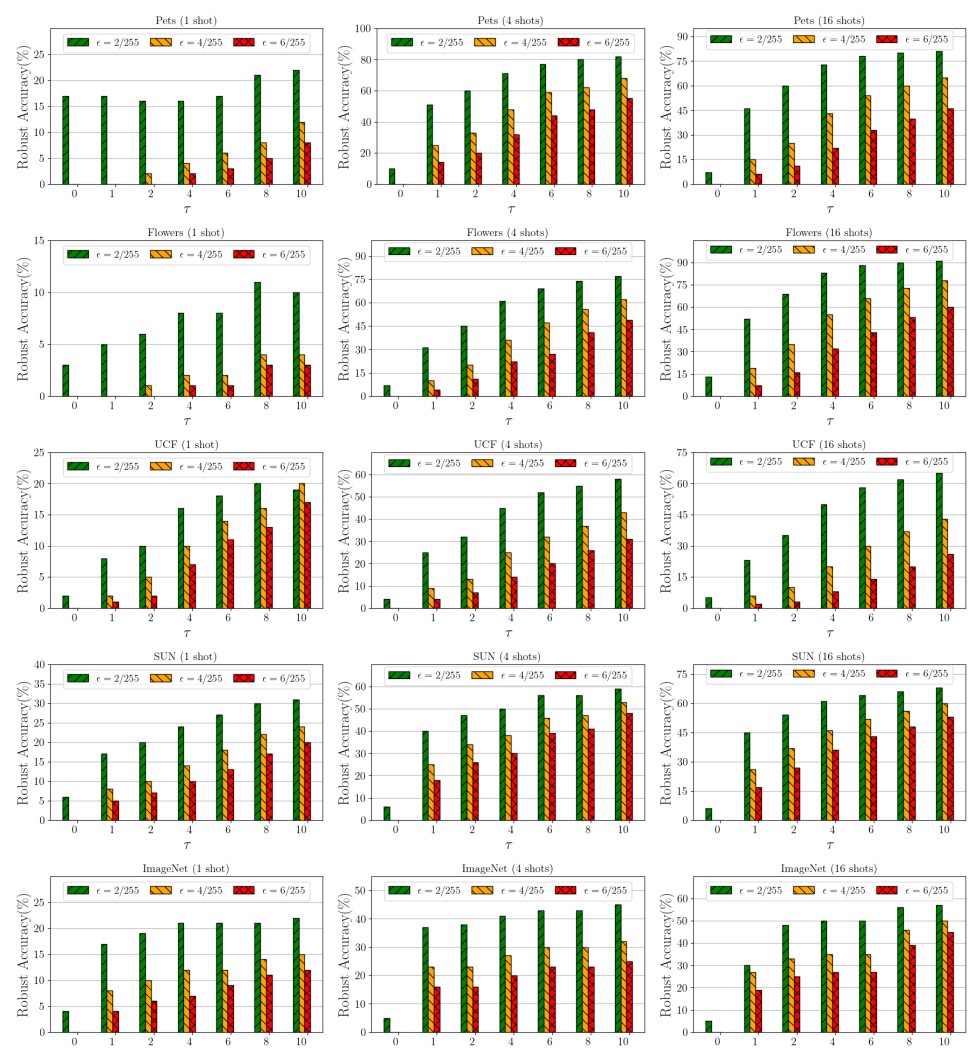

Figure 6: Robust accuracy of AdvCLIP-LoRA with ViT-B/16 backbone on Pets, Flowers, UCF, and SUN datasets with different $\tau$ and $\epsilon$ values.

### B.4 ABLATION ON LORA DESIGN CHOICES

Table 8: Average Clean, PGD-100, and harmonic mean (HM) for LoRA variants.

| Method | Overall Average | | | Flowers | | | Pets | | | SUN | | | UCF | | |
|---|---|---|---|---|---|---|---|---|---|---|---|---|---|---|---|
| | Clean | PGD | HM | Clean | PGD | HM | Clean | PGD | HM | Clean | PGD | HM | Clean | PGD | HM |
| AdvCLIP-LoRA | 81.25 | 34.76 | 48.69 | 90.70 | 48.72 | 63.39 | 87.03 | 32.98 | 47.83 | 71.09 | 31.11 | 43.28 | 76.18 | 26.22 | 39.01 |
| Vision | 78.71 | 30.74 | 44.21 | 86.07 | 37.72 | 52.45 | 88.74 | 35.08 | 50.28 | 67.40 | 26.02 | 37.55 | 72.64 | 24.13 | 36.23 |
| $W_q$ | 80.65 | 30.62 | 44.39 | 87.86 | 37.07 | 52.14 | 88.09 | 33.69 | 48.74 | 70.88 | 28.34 | 40.49 | 75.76 | 23.39 | 35.74 |
| $W_v$ | 80.95 | 34.73 | 48.61 | 89.05 | 45.43 | 60.17 | 87.14 | 35.62 | 50.57 | 70.53 | 31.18 | 43.24 | 77.09 | 26.40 | 39.33 |
| $W_q W_v$ | 80.95 | 34.65 | 48.53 | 89.85 | 48.96 | 63.38 | 86.86 | 33.69 | 48.55 | 71.22 | 30.44 | 42.65 | 75.87 | 25.51 | 38.18 |
| up | 81.21 | 29.32 | 43.08 | 90.17 | 38.81 | 54.26 | 88.42 | 30.25 | 45.08 | 70.35 | 25.95 | 37.91 | 75.89 | 22.28 | 34.45 |
| bottom | 80.09 | 33.02 | 46.76 | 88.10 | 41.18 | 56.13 | 87.03 | 36.77 | 51.70 | 70.30 | 31.11 | 43.13 | 74.91 | 23.00 | 35.19 |
| half-up | 81.37 | 30.72 | 44.60 | 90.05 | 41.53 | 56.84 | 88.14 | 29.82 | 44.56 | 70.40 | 27.14 | 39.18 | 76.90 | 24.40 | 37.05 |
| half-bottom | 79.80 | 32.70 | 46.39 | 88.79 | 42.43 | 57.42 | 85.55 | 33.52 | 48.17 | 70.33 | 31.26 | 43.28 | 74.52 | 23.61 | 35.86 |
| mid | 80.45 | 30.98 | 44.73 | 87.82 | 39.95 | 54.92 | 88.31 | 32.38 | 47.39 | 69.92 | 28.40 | 40.39 | 75.73 | 23.18 | 35.50 |

## C  CONVERGENCE ANALYSIS

Before presenting the main theorem, we state several key intermediate lemmas used in the proof. For notational convenience, we denote $\Phi(W := W_0 + BA)$ as $\Phi(BA)$, and use $\Phi(W)$ and $\Phi(BA)$ interchangeably throughout the analysis. Let us begin with a few definitions.

**Definition C.1** *A function $f$ is L-Lipschitz if for all $W, W'$, we have*

$$\|f(W) - f(W')\| \le L \|W - W'\|. \tag{12}$$

**Definition C.2** *A function $f$ is $\ell$-smooth if for all $W, W'$, we have*

$$\|\nabla f(W) - \nabla f(W')\| \le \ell \|W - W'\|. \tag{13}$$

**Proposition C.1** *Lin et al. (2020) Under Assumption 4.2, $\Phi(\cdot)$ is $2\kappa\ell$-smooth with $\nabla\Phi(\cdot) = \nabla_W f(\cdot, \delta^\star(\cdot))$. Also, $\delta^\star(\cdot)$ is $\kappa$-Lipschitz.*

**Lemma C.1** *For any matrices $A, B \in \mathbb{R}^{d \times k}$ and $\alpha, \delta > 0$ we have*

$$2\langle A, B \rangle \le \delta \|A\|^2 + \delta^{-1} \|B\|^2,$$
$$\|A + B\|^2 \le (1 + \alpha)\|A\|^2 + (1 + \tfrac{1}{\alpha})\|B\|^2. \tag{14}$$

Using Proposition C.1 and $\|A\|_F \le c_A$, $\|B\|_F \le c_B$, we can prove the smoothness of $\Phi(\cdot)$ with respect to $A$ and $B$ when the other is held fixed. Formally, we state the following lemma:

**Lemma C.2** *Under Assumption 4.2 and boundedness of low-rank matrices, the function $\Phi$ is $2\kappa\ell c_B^2$-smooth with respect to $A$ when $B$ is fixed, and $2\kappa\ell c_A^2$-smooth with respect to $B$ when $A$ is fixed.*

*Proof.* First, by the chain rule we notice that

$$\nabla_A \Phi(W) = \nabla_A f(W, \delta^*(W)) = B^T \nabla_W f(W, \delta^*(W)) + \left(\frac{\mathrm{d}\delta^*(W)}{\mathrm{d}W}\right)^T \underbrace{\nabla_\delta f(W, \delta^*(W))}_{=0}$$
$$= B^T \nabla_W \Phi(W). \tag{15}$$

Similarly, we have:

$$\nabla_B \Phi(W) = \nabla_W \Phi(W) A^T. \tag{16}$$

Now, we can write

$$
\begin{aligned}
\|\nabla_A \Phi(BA) - \nabla_A \Phi(BA')\| &= \left\| B^T \nabla_W \Phi(BA) - B^T \nabla_W \Phi(BA') \right\| \\
&= \|B\| \, \|\nabla_W \Phi(BA) - \nabla_W \Phi(BA')\| \\
&\overset{(a)}{\le} c_B(2\kappa\ell) \, \|BA - BA'\| \\
&\le 2\kappa\ell c_B^2 \, \|A - A'\|.
\end{aligned}
\tag{17}
$$

In $(a)$, we used the boundedness of the low-rank matrices and Proposition C.1. Similarly, we can prove that $\Phi$ is $2\kappa\ell c_A^2$-smooth with respect to $B$ when $A$ is fixed. $\qquad\square$

**Lemma C.3** *The iterates $\{A_t, B_t\}_{t \ge 1}$ in Alg. 1 (lines 8-9) satisfy the following inequality:*

$$
\begin{aligned}
\mathbb{E}\Phi(B_t A_t) \le\ & \mathbb{E}\Phi(B_{t-1} A_{t-1}) - \tfrac{\eta_w}{2}\left( \mathbb{E}\|\nabla_A \Phi(B_{t-1} A_{t-1})\|^2 + \mathbb{E}\|\nabla_B \Phi(B_{t-1} A_{t-1})\|^2 \right) \\
& + \tfrac{5\eta_w}{4} \mathbb{E}\|\nabla_A f(B_{t-1} A_{t-1}, \delta_t) - \nabla_A \Phi(B_{t-1} A_{t-1})\|^2 \\
& + \tfrac{\eta_w}{2} \mathbb{E}\|\nabla_B f(B_{t-1} A_{t-1}, \delta_t) - \nabla_B \Phi(B_{t-1} A_{t-1})\|^2 \\
& + \tfrac{\kappa\ell(c_A^4 + c_B^4)\eta_w^2 G^2}{M} + \tfrac{2G^2(2\kappa\ell c_B^2 c_A^4 + G^2)\eta_w^3}{M}.
\end{aligned}
\tag{18}
$$

*Proof.* Using smoothness for $A$ from Lemma C.2, we can write

$$\mathbb{E}\Phi(B_t A_t) \leq \mathbb{E}\Phi(B_t A_{t-1}) + \mathbb{E}\langle\nabla_A\Phi(B_t A_{t-1}), A_t - A_{t-1}\rangle + \kappa\ell c_B^2\eta_w^2\mathbb{E}\|A_t - A_{t-1}\|^2$$

$$\leq \mathbb{E}\Phi(B_t A_{t-1}) + \mathbb{E}\langle\nabla_A\Phi(B_t A_{t-1}), -\eta_w\nabla_A f(B_{t-1}A_{t-1}, \delta_t)\rangle$$

$$+ \kappa\ell c_B^2\eta_w^2\mathbb{E}\left\|\frac{1}{M}\sum_{i=1}^M \nabla_A F(B_{t-1}A_{t-1}, \delta_t; \xi_i)\right\|^2$$

$$\overset{(a)}{\leq} \mathbb{E}\Phi(B_t A_{t-1}) + \frac{\kappa\ell c_B^4\eta_w^2 G^2}{M}$$

$$+ \mathbb{E}\langle\nabla_A\Phi(B_t A_{t-1}) - \nabla_A\Phi(B_{t-1}A_{t-1}) + \nabla_A\Phi(B_{t-1}A_{t-1}), -\eta_w\nabla_A f(B_{t-1}A_{t-1}, \delta_t)\rangle$$

$$= \mathbb{E}\Phi(B_t A_{t-1}) - \eta_w\mathbb{E}\langle\nabla_A\Phi(B_t A_{t-1}) - \nabla_A\Phi(B_{t-1}A_{t-1}), \nabla_A f(B_{t-1}A_{t-1}, \delta_t)\rangle$$

$$- \eta_w\mathbb{E}\langle\nabla_A\Phi(B_{t-1}A_{t-1}), \nabla_A f(B_{t-1}A_{t-1}, \delta_t)\rangle + \frac{\kappa\ell c_B^4\eta_w^2 G^2}{M}$$

$$\overset{(b)}{\leq} \mathbb{E}\Phi(B_t A_{t-1}) + 2\eta_w\mathbb{E}\|\nabla_A\Phi(B_t A_{t-1}) - \nabla_A\Phi(B_{t-1}A_{t-1})\|^2 + \frac{\eta_w}{8}\mathbb{E}\|\nabla_A f(B_{t-1}A_{t-1}, \delta_t)\|^2$$

$$- \eta_w\mathbb{E}\langle\nabla_A\Phi(B_{t-1}A_{t-1}), \nabla_A f(B_{t-1}A_{t-1}, \delta_t) - \nabla_A\Phi(B_{t-1}A_{t-1}) + \nabla_A\Phi(B_{t-1}A_{t-1})\rangle$$

$$+ \frac{\kappa\ell c_B^4\eta_w^2 G^2}{M}$$

$$\overset{(c)}{\leq} \mathbb{E}\Phi(B_t A_{t-1}) + 2\eta_w\mathbb{E}\|\nabla_A\Phi(B_t A_{t-1}) - \nabla_A\Phi(B_{t-1}A_{t-1})\|^2 + \frac{\eta_w}{4}\mathbb{E}\|\nabla_A\Phi(B_{t-1}A_{t-1})\|^2$$

$$+ \frac{\eta_w}{4}\mathbb{E}\|\nabla_A\Phi(B_{t-1}A_{t-1}) - \nabla_A f(B_{t-1}A_{t-1}, \delta_t)\|^2 - \frac{3\eta_w}{4}\mathbb{E}\|\nabla_A\Phi(B_{t-1}A_{t-1})\|^2$$

$$+ \eta_w\mathbb{E}\|\nabla_A f(B_{t-1}A_{t-1}, \delta_t) - \nabla_A\Phi(B_{t-1}A_{t-1})\|^2 + \frac{\kappa\ell c_B^4\eta_w^2 G^2}{M}$$

$$= \mathbb{E}\Phi(B_t A_{t-1}) + 2\eta_w\mathbb{E}\|\nabla_A\Phi(B_t A_{t-1}) - \nabla_A\Phi(B_{t-1}A_{t-1})\|^2 - \frac{\eta_w}{2}\mathbb{E}\|\nabla_A\Phi(B_{t-1}A_{t-1})\|^2$$

$$+ \frac{5\eta_w}{4}\mathbb{E}\|\nabla_A f(B_{t-1}A_{t-1}, \delta_t) - \nabla_A\Phi(B_{t-1}A_{t-1})\|^2 + \frac{\kappa\ell c_B^4\eta_w^2 G^2}{M}. \tag{19}$$

In $(a)$ we applied Assumption 4.1, in $(b)$ we employed the inequality $\langle a, b\rangle \leq \frac{1}{8}\|a\|^2 + 2\|b\|^2$, and in $(c)$ we utilized the inequalities $\langle a, b\rangle \leq \frac{1}{4}\|a\|^2 + \|b\|^2$ and $\|a + b\|^2 \leq 2\|a\|^2 + 2\|b\|^2$. We derive the following bound on the term in the above inequality:

$$\mathbb{E}\|\nabla_A\Phi(B_t A_{t-1}) - \nabla_A\Phi(B_{t-1}A_{t-1})\|^2 \leq \mathbb{E}\|B_t^T\nabla_W\Phi(B_t A_{t-1}) - B_{t-1}^T\nabla_W\Phi(B_{t-1}A_{t-1})\|^2$$

$$\leq \mathbb{E}\|B_t^T\nabla_W\Phi(B_t A_{t-1}) - B_t^T\nabla_W\Phi(B_{t-1}A_{t-1})\|^2$$

$$+ \mathbb{E}\|B_t^T\nabla_W\Phi(B_{t-1}A_{t-1}) - B_{t-1}^T\nabla_W\Phi(B_{t-1}A_{t-1})\|^2$$

$$\leq 2\kappa\ell c_B^2 c_A^2\mathbb{E}\|B_t - B_{t-1}\|^2 + \mathbb{E}\|B_t^T - B_{t-1}^T\|^2 G^2$$

$$\leq \frac{2\kappa\ell c_B^2 c_A^4 G^2\eta_w^2}{M} + \frac{G^4\eta_w^2}{M}. \tag{20}$$

If we use equation 20 in equation 19, we have

$$\mathbb{E}\Phi(B_t A_t) \leq \mathbb{E}\Phi(B_t A_{t-1}) - \frac{\eta_w}{2}\mathbb{E}\|\nabla_A\Phi(B_{t-1}A_{t-1})\|^2$$

$$+ \frac{5\eta}{4}\mathbb{E}\|\nabla_A f(B_{t-1}A_{t-1}, \delta_t) - \nabla_A\Phi(B_{t-1}A_{t-1})\|^2$$

$$+ \frac{\kappa\ell c_B^4\eta_w^2 G^2}{M} + \frac{4\kappa\ell c_B^2 c_A^4 G^2\eta_w^3}{M} + \frac{2G^4\eta_w^3}{M}. \tag{21}$$

Using smoothness for $B$ from Lemma C.2, we can write

$$\mathbb{E}\Phi(B_t A_{t-1}) \leq \mathbb{E}\Phi(B_{t-1}A_{t-1}) + \mathbb{E}\langle\nabla_B\Phi(B_{t-1}A_{t-1}), B_t - B_{t-1}\rangle + \kappa\ell c_A^2\eta_w^2\mathbb{E}\|B_t - B_{t-1}\|^2$$

$$\leq \mathbb{E}\Phi(B_{t-1}A_{t-1}) + \mathbb{E}\langle\nabla_B\Phi(B_{t-1}A_{t-1}), -\eta_w\nabla_B f(B_{t-1}A_{t-1}, \delta_t)\rangle$$

$$+ \kappa\ell c_A^2\eta_w^2\mathbb{E}\left\|\frac{1}{M}\sum_{i=1}^M \nabla_B F(B_{t-1}A_{t-1}, \delta_t; \xi)\right\|^2$$

$$\leq \mathbb{E}\Phi(B_{t-1}A_{t-1}) + \frac{\kappa\ell c_A^4\eta_w^2 G^2}{M}$$

$$- \eta_w\mathbb{E}\langle\nabla_B\Phi(B_{t-1}A_{t-1}), \nabla_B f(B_{t-1}A_{t-1}, \delta_t) - \nabla_B\Phi(B_{t-1}A_{t-1}) + \nabla_B\Phi(B_{t-1}A_{t-1})\rangle$$

$$\leq \mathbb{E}\Phi(B_{t-1}A_{t-1}) - \frac{\eta_w}{2}\mathbb{E}\|\nabla_B\Phi(B_{t-1}A_{t-1})\|^2 + \frac{\kappa\ell c_A^4\eta_w^2 G^2}{M}$$

$$+ \frac{\eta_w}{2}\mathbb{E}\|\nabla_B f(B_{t-1}A_{t-1}, \delta_t) - \nabla_B\Phi(B_{t-1}A_{t-1})\|^2. \tag{22}$$

Summing equation 21 and equation 22 yields the desired inequality. $\qquad\square$

**Lemma C.4** *Let* $\gamma_t = \mathbb{E}\left\|\delta^\star(W_t) - \delta_t\right\|^2$, *the following statement holds true,*

$$\gamma_t \leq \left(1 - \tfrac{1}{2\kappa}\right)\gamma_{t-1} + \tfrac{8\kappa^3(c_A^4 + c_B^4)G^2\eta_w^2}{M} + \tfrac{2G^2}{\ell^2 M}. \tag{23}$$

*Proof.* Since $f(W_t, \cdot)$ is $\mu$-strongly concave and $\eta_\delta = 1/\ell$, we have Lin et al. (2020)

$$\mathbb{E}\left\|\delta^\star(W_{t-1}) - \delta_t\right\|^2 \leq \left(1 - \tfrac{1}{\kappa}\right)\gamma_{t-1} + \tfrac{2G^2}{\ell^2 M}. \tag{24}$$

We can also write

$$\gamma_t \leq \left(1 + \tfrac{1}{2(\max\{\kappa, 2\} - 1)}\right)\mathbb{E}\left\|\delta^\star(W_{t-1}) - \delta_t\right\|^2$$
$$+ (1 + 2(\max\{\kappa, 2\} - 1))\mathbb{E}\left\|\delta^\star(W_t) - \delta^\star(W_{t-1})\right\|^2$$
$$\leq \left(\tfrac{2\max\{\kappa, 2\} - 1}{2\max\{\kappa, 2\} - 2}\right)\mathbb{E}\left\|\delta^\star(W_{t-1}) - \delta_t\right\|^2 + 4\kappa\mathbb{E}\left\|\delta^\star(W_t) - \delta^\star(W_{t-1})\right\|^2$$
$$\overset{(a)}{\leq} \left(1 - \tfrac{1}{2\kappa}\right)\gamma_{t-1} + 4\kappa\mathbb{E}\left\|\delta^\star(W_t) - \delta^\star(W_{t-1})\right\|^2 + \tfrac{2G^2}{\ell^2 M}, \tag{25}$$

where in $(a)$ we used equation 24. Since $\delta^\star(\cdot)$ is $\kappa$-Lipschitz, $\left\|\delta^\star(W_t) - \delta^\star(W_{t-1})\right\| \leq \kappa\left\|W_t - W_{t-1}\right\|$. Furthermore, we have

$$\mathbb{E}\left\|W_t - W_{t-1}\right\|^2 = \mathbb{E}\left\|B_t A_t - B_t A_{t-1} + B_t A_{t-1} - B_{t-1}A_{t-1}\right\|^2$$
$$\leq 2c_B^2\mathbb{E}\left\|A_t - A_{t-1}\right\|^2 + 2c_A^2\mathbb{E}\left\|B_t - B_{t-1}\right\|^2$$
$$= \tfrac{2G^2(c_A^4 + c_B^4)\eta_w^2}{M}. \tag{26}$$

Using equation 26 into equation 25 yields the desired inequality $\qquad\square$

**Lemma C.5** *Let* $\gamma_t = \mathbb{E}\left\|\delta^\star(W_t) - \delta_t\right\|^2$, *the following statement holds true,*

$$\mathbb{E}\Phi(B_t A_t) \leq \mathbb{E}\Phi(B_{t-1}A_{t-1}) - \tfrac{\eta_w}{2}\left(\mathbb{E}\left\|\nabla_A\Phi(B_{t-1}A_{t-1})\right\|^2 + \mathbb{E}\left\|\nabla_B\Phi(B_{t-1}A_{t-1})\right\|^2\right)$$
$$+ \ell^2\eta_w\left(\tfrac{5c_B^2 + 2c_A^2}{2}\right)\gamma_{t-1} + \tfrac{G^2(2.5c_B^2 + c_A^2)\eta_w}{M} + \tfrac{\kappa\ell(c_A^4 + c_B^4)G^2\eta_w^2}{M} + \tfrac{2G^2(2\kappa\ell c_B^2 c_A^4 + G^2)\eta_w^3}{M}. \tag{27}$$

*Proof.* Since $\nabla_W\Phi(W_{t-1}) = \nabla_W f(W_{t-1}, \delta^*(W_{t-1}))$, we have

$$\mathbb{E}\left\|\nabla_A f(W_{t-1}, \delta^*(W_{t-1})) - \nabla_A f(W_{t-1}, \delta_t)\right\|^2$$
$$= \mathbb{E}\left\|B_{t-1}^T\nabla_A f(W_{t-1}, \delta^*(W_{t-1})) - B_{t-1}^T\nabla_A f(W_{t-1}, \delta_t)\right\|^2$$
$$\leq c_B^2\ell^2\mathbb{E}\left\|\delta^*(W_{t-1}) - \delta_t\right\|^2 \leq 2c_B^2\ell^2\left(\mathbb{E}\left\|\delta^*(W_{t-1}) - \delta_{t-1}\right\|^2 + \mathbb{E}\left\|\delta_t - \delta_{t-1}\right\|^2\right)$$
$$\leq 2c_B^2\ell^2\left(\gamma_{t-1} + \tfrac{G^2}{\ell^2 M}\right) = 2c_B^2\ell^2\gamma_{t-1} + \tfrac{2c_B^2 G^2}{M}. \tag{28}$$

Similarly, we have

$$\mathbb{E}\left\|\nabla_B f(W_{t-1}, \delta^*(W_{t-1})) - \nabla_B f(W_{t-1}, \delta_t)\right\|^2 \leq 2c_A^2\ell^2\gamma_{t-1} + \tfrac{2c_A^2 G^2}{M}. \tag{29}$$

Combining equation 28 and equation 29 with equation 18 yields the desired inequality. $\qquad\square$

**Theorem C.1** *Let Assumptions 4.1 and 4.2 hold. Moreover, assume that the low-rank matrices remain bounded by constants $c_A$ and $c_B$ in each iteration, i.e., $\|A_t\|_F \leq c_A$ and $\|B_t\|_F \leq c_B$. Then, there exists iteration $t \in \{0, \cdots, T - 1\}$ for which*

$$\mathbb{E}\left\|\nabla\Phi(W_t)\right\|^2 \leq \mathcal{O}\left(\tfrac{4\Delta_\Phi(1/\eta_w) + \kappa\ell^2(c_A^2 + c_B^2)D^2}{\epsilon^2}\right), \tag{30}$$

*where $\eta_w = \Theta(\min\{1/\kappa\ell(c_A^4 + c_B^4), 1/\kappa^2\ell(c_A^2 + c_B^2), 1/(G^2 + \kappa\ell c_A^4 c_B^2)^{1/2}\})$, $\eta_\delta = \Theta(1/\ell)$, and $\Delta_\Phi = \mathbb{E}\Phi(W_0) - \mathbb{E}\Phi(W_{T+1})$. Moreover, the mini-batch size $M$ is bounded by*

$$\mathcal{O}\left(\tfrac{G^2 + \kappa(c_A^2 + c_B^2)G^2}{\epsilon^2}\right). \tag{31}$$

*Proof.* Performing the inequality in Lemma C.4 recursively and using $\gamma_0 \leq D^2$ from Assumption 4.2 results in

$$\gamma_t \leq \left(1 - \tfrac{1}{2\kappa}\right)^t D^2 + \left(\tfrac{8\kappa^3(c_A^4 + c_B^4)G^2\eta_w^2}{M} + \tfrac{2G^2}{\ell^2 M}\right)\left(\sum_{j=0}^{t-1}\left(1 - \tfrac{1}{2\kappa}\right)^{t-1-j}\right). \tag{32}$$

Combining equation 32 with equation 27, we have

$$\mathbb{E}\Phi(W_t) \leq \mathbb{E}\Phi(W_{t-1}) - \tfrac{\eta_w}{2}\left(\mathbb{E}\left\|\nabla_A\Phi(W_{t-1})\right\|^2 + \mathbb{E}\left\|\nabla_B\Phi(W_{t-1})\right\|^2\right)$$

$$+ \eta_w\ell^2\left(\tfrac{5c_B^2 + 2c_A^2}{2}\right)\left(1 - \tfrac{1}{2\kappa}\right)^{t-1}D^2$$

$$+ \eta_w\ell^2\left(\tfrac{5c_B^2 + 2c_A^2}{2}\right)\left(\tfrac{8\kappa^3(c_A^4 + c_B^4)G^2\eta_w^2}{M} + \tfrac{2G^2}{\ell^2 M}\right)\left(\sum_{j=0}^{t-2}\left(1 - \tfrac{1}{2\kappa}\right)^{t-2-j}\right)$$

$$+ \tfrac{G^2(2.5c_B^2 + c_A^2)\eta_w}{M} + \tfrac{\kappa\ell(c_A^4 + c_B^4)G^2\eta_w^2}{M} + \tfrac{2G^2(2\kappa\ell c_B^2 c_A^4 + G^2)\eta_w^3}{M}. \tag{33}$$

Summing up equation 33 over $t = 1, 2, \cdots, T+1$ and rearranging, we can write

$$\mathbb{E}\Phi(W_{T+1}) \leq \mathbb{E}\Phi(W_0) - \tfrac{\eta_w}{2}\sum_{t=0}^{T}\left(\mathbb{E}\left\|\nabla_A\Phi(W_t)\right\|^2 + \mathbb{E}\left\|\nabla_B\Phi(W_t)\right\|^2\right)$$

$$+ \eta_w\ell^2\left(\tfrac{5c_B^2 + 2c_A^2}{2}\right)D^2\left(\sum_{t=0}^{T}\left(1 - \tfrac{1}{2\kappa}\right)^t\right)$$

$$+ \eta_w\ell^2\left(\tfrac{5c_B^2 + 2c_A^2}{2}\right)\left(\tfrac{8\kappa^3(c_A^4 + c_B^4)G^2\eta_w^2}{M} + \tfrac{2G^2}{\ell^2 M}\right)\left(\sum_{t=1}^{T+1}\sum_{j=0}^{t-2}\left(1 - \tfrac{1}{2\kappa}\right)^{t-2-j}\right)$$

$$+ \tfrac{G^2(2.5c_B^2 + c_A^2)\eta_w(T+1)}{M} + \tfrac{\kappa\ell(c_A^4 + c_B^4)G^2\eta_w^2(T+1)}{M} + \tfrac{2G^2(2\kappa\ell c_B^2 c_A^4 + G^2)\eta_w^3(T+1)}{M}$$

$$\leq \mathbb{E}\Phi(W_0) - \tfrac{\eta_w}{2}\sum_{t=0}^{T}\left(\mathbb{E}\left\|\nabla_A\Phi(W_t)\right\|^2 + \mathbb{E}\left\|\nabla_B\Phi(W_t)\right\|^2\right) + \kappa\eta_w\ell^2\left(5c_B^2 + 2c_A^2\right)D^2$$

$$+ \kappa\eta_w\ell^2\left(5c_B^2 + 2c_A^2\right)\left(\tfrac{8\kappa^3(c_A^4 + c_B^4)G^2\eta_w^2}{M} + \tfrac{2G^2}{\ell^2 M}\right)(T+1)$$

$$+ \tfrac{G^2(2.5c_B^2 + c_A^2)\eta_w(T+1)}{M} + \tfrac{\kappa\ell(c_A^4 + c_B^4)G^2\eta_w^2(T+1)}{M} + \tfrac{2G^2(2\kappa\ell c_B^2 c_A^4 + G^2)\eta_w^3(T+1)}{M}. \tag{34}$$

Then, it follows that

$$\tfrac{1}{T+1}\sum_{t=0}^{T}\mathbb{E}\left\|\nabla_{(A,B)}\Phi(W_t)\right\|^2 = \tfrac{1}{T+1}\sum_{t=0}^{T}\left(\mathbb{E}\left\|\nabla_A\Phi(W_t)\right\|^2 + \mathbb{E}\left\|\nabla_B\Phi(W_t)\right\|^2\right) \leq \tfrac{2(\mathbb{E}\Phi(W_0) - \mathbb{E}\Phi(W_{T+1}))}{\eta_w(T+1)}$$

$$+ \tfrac{\kappa\ell^2\left(10c_B^2 + 4c_A^2\right)D^2}{T+1} + \kappa\ell^2\left(10c_B^2 + 4c_A^2\right)\left(\tfrac{8\kappa^3(c_A^4 + c_B^4)G^2\eta_w^2}{M} + \tfrac{2G^2}{\ell^2 M}\right) + \tfrac{2G^2(2.5c_B^2 + c_A^2)}{M}$$

$$+ \tfrac{2\kappa\ell(c_A^4 + c_B^4)G^2\eta_w}{M} + \tfrac{4G^2(2\kappa\ell c_B^2 c_A^4 + G^2)\eta_w^2}{M}$$

$$\leq \mathcal{O}\left(\tfrac{\Delta_\Phi}{\eta_w(T+1)} + \tfrac{\kappa\ell^2(c_A^2 + c_B^2)D^2}{T+1} + \tfrac{G^2}{M} + \tfrac{\kappa(c_A^2 + c_B^2)G^2}{M}\right). \tag{35}$$

This implies that the number of iterations required by Algorithm 1 to return an $\epsilon$-stationary point is bounded by

$$\mathcal{O}\left(\frac{4\Delta_\Phi(1/\eta_w) + \kappa\ell^2(c_A^2 + c_B^2)D^2}{\epsilon^2}\right), \tag{36}$$

Moreover, the mini-batch size $M$ is bounded by

$$\mathcal{O}\left(\frac{G^2 + \kappa(c_A^2 + c_B^2)G^2}{\epsilon^2}\right), \tag{37}$$

which completes the proof. $\qquad\square$

