# OpenReview forum: "Few-Shot Adversarial Low-Rank Fine-Tuning of Vision-Language Models"
_ICLR.cc/2026/Conference — ICLR 2026 Conference Withdrawn Submission_

### Official Review · Reviewer_jvcV · 2025-10-31

**Soundness:** 2
**Presentation:** 3
**Contribution:** 2
**Rating:** 4
**Confidence:** 4

**Summary:**

The paper specifically focuses on adapting Parameter-Efficient Fine-Tuning (PEFT) techniques (e.g., LoRA) to adversarial VLM finetuning to enhance robustness in an efficient scheme. The paper also introduces convergence analyses of the proposed AdvCLIP-LoRA method from a theoretical perspective. Extensive experiments across diverse datasets and backbones demonstrate the efficacy of the proposed method. In addition to adversarially robust few-shot learning, the paper also investigated zero-shot robustness transfer to other unforeseen datasets. Ablation analyses across diverse setups/design choices demonstrate the efficacy of the proposed method.

**Strengths:**

1. The paper is well-written and organized. The storyline is straightforward and intuitive. The methodology part is easy to follow.
2. The paper introduces theoretical analyses regarding the convergence of the proposed PEFT-based adversarial finetuning method.
3. According to experimental results (especially Tables 1&2), the proposed method shows a significant improvement beyond previous approaches.

**Weaknesses:**

1. The proposed method seems to be a simple combination of LoRA and standard adversarial fine-tuning (TeCoA) [a]. I find it hard to get some novel insights from the paper. Can authors explicitly show the difficulty of merging these two techniques?

2. In addition, the paper lacks experimental comparisons with previous adversarial fine-tuning approaches [b,c].

3. In addition to LoRA, there also exist a lot of PEFT strategies; the author can also discuss and compare some of them, e.g., vision prompt tuning.

4. The paper primarily focuses on robustness evaluation of PGD attacks. However, a lot of adaptive and composite adversarial attack methods are missing, e.g., AutoAttack. The paper should also explore diverse downstream visual-language task extensions.

5. Training time cost comparisons and complexity analyses are missing.

[a] Understanding zero-shot adversarial robustness for large-scale models (ICLR 2023)
[b] Pre-trained model guided fine-tuning for zero-shot adversarial robustness (CVPR 2024)
[c] Robust clip: Unsupervised adversarial fine-tuning of vision embeddings for robust large vision-language models (ICML 2024)

**Questions:**

1. The paper should include additional comparisons with other PEFT strategies.
2. Can the authors also test hallucination scenarios in comparison with adversarial perturbations?
3. Would there be any cross-domain cases for robust few-shot learning?
4. In addition to the vanilla LoRA extension, is it possible to test some recently improved LoRA pipelines?

---

### Official Review · Reviewer_MkaV · 2025-10-31

**Soundness:** 2
**Presentation:** 2
**Contribution:** 2
**Rating:** 4
**Confidence:** 3

**Summary:**

This paper introduces AdvCLIP-LoRA, a parameter-efficient adversarial fine-tuning framework that integrates LoRA into CLIP for few-shot learning scenarios. By formulating the problem as a minimax optimization between adversarial perturbations and low-rank adapter parameters, the method aims to improve both robustness and efficiency. Experimental results across multiple datasets demonstrate that AdvCLIP-LoRA achieves strong clean and adversarial accuracy.

**Strengths:**

- The paper presents a systematic exploration of adversarial robustness in few-shot LoRA-based CLIP adaptation, offering a parameter-efficient alternative to adversarial prompt tuning that achieves superior results on both clean and robust accuracy.
- The paper provides a nontrivial convergence guarantee for the minimax optimization, enhancing the methodological soundness and theoretical completeness of the work.

**Weaknesses:**

- The paper lacks depth. It mainly demonstrates the numerical advantages of LoRA-based adversarial fine-tuning without providing a clear explanation of why it performs better than prompt-tuning-based methods. As an attempt to apply adversarial fine-tuning to a new adaptation paradigm, the paper should comprehensively compare and discuss various fine-tuning frameworks (e.g., prompt tuning, adapter tuning, full fine-tuning) to strengthen its contribution.
- The experiments are not sufficiently comprehensive, as they only evaluate performance under PGD attacks without including other attack types (e.g., AutoAttack, CW, or black-box attacks).
- The paper lacks runtime and memory cost analysis, as well as a direct comparison of computational efficiency between LoRA-based and prompt tuning-based adversarial fine-tuning methods.

**Questions:**

Please refer to the weaknesses.

---

### Official Review · Reviewer_r64Y · 2025-11-01

**Soundness:** 3
**Presentation:** 3
**Contribution:** 2
**Rating:** 4
**Confidence:** 4

**Summary:**

This paper proposes AdvCLIP-LoRA, a method for enhancing the adversarial robustness of Vision-Language Models (CLIP) in the few-shot learning setting. The authors claim to be the first to explore the LoRA (Low-Rank Adaptation) and Adversarial Training for few-shot VLM adversarial robustness, in contrast to the prompt-tuning based adversarial methods. The paper also provides a theoretical convergence analysis that guarantees convergence to a stationary point.

**Strengths:**

1.	The application of combining LoRA with adversarial training for adversarial robustness in few-shot VLMs.
2.	Excellent experimental performance, especially compared to prompt-tuning methods in the 1-shot setting.

**Weaknesses:**

1.	Limited novelty. Related methods have already been studied in [1, 2, 3], e.g., [1] investigated a similar method, though not specifically in the few-shot setting.
2.	Limited theoretical contribution. The convergence analysis relies on several strong assumptions, such as the guarantee that the low-rank matrices A and B remain bounded in each iteration (i.e., not exceeding constants cA and cB in line 286). However, the optimization process lacks explicit constraint on the matrix norms. Therefore, the convergence proof is merely an application of standard non-convex optimization problems and may not provide valuable insights for practice.
3.	Few-shot VLMs emphasize data-scarce, resource-constrained settings, however, adversarial training is computationally expensive. The paper lacks discussion on the trade-off between training cost and performance, such as a comparison of training time with [4].
4.	Experimental Issues:
A. There are significant discrepancies between Table 1 and the original paper [3], particularly in the experimental data for 5 methods (AdvVP, APT, AdvMaPLe, AdvVLP, FAP) on OxfordPets, Food101, SUN397.
B. Related papers and follow-up works often use 11 datasets. Why were EuroSAT, FGVCAircraft, and StanfordCars not included in the experiments?
C. In Figure 4 (Top-right), the ablation on the attack budget ϵ is incomplete. Would using ϵ=1/255 or 3/255 yield better results?
D. The performance of the text-only is missing in Table 4.
E. Some experimental data matches the original paper exactly, while others show minor discrepancies (e.g., AdvVLP in the 1-shot setting on ImageNet-1K in Table 1). Were these methods reproduced? Please provide clarification.

[1] Ji, Yuheng, et al. "Advlora: Adversarial low-rank adaptation of vision-language models." (2024).

[2] Zanella, Maxime, and Ismail Ben Ayed. "Low-rank few-shot adaptation of vision-language models." Proceedings of the IEEE/CVF Conference on Computer Vision and Pattern Recognition. 2024.

[3] Yuan Z, Zhang J, Shan S, et al. FullLoRA: Efficiently Boosting the Robustness of Pretrained Vision Transformers[J]. IEEE Transactions on Image Processing, 2025.

[4] Zhou, Yiwei, et al. "Few-shot adversarial prompt learning on vision-language models." Advances in Neural Information Processing Systems 37 (2024): 3122-3156.

**Questions:**

My major concerns lie in the limiations on novelty and experiments. See weaknesses for details.

---

### Official Review · Reviewer_Sya2 · 2025-11-01

**Soundness:** 3
**Presentation:** 3
**Contribution:** 2
**Rating:** 4
**Confidence:** 4

**Summary:**

The paper proposes a parameter-efficient adversarial fine-tuning scheme (AdvCLIP-LoRA) that applies LoRA to both CLIP encoders and jointly optimizes low-rank adapters with inner-loop PGD on images (generating adversarial images) under a minimax objective. Experiments show the consistent gains in adversarial robustness with minial natural performance drop across different datasets and backbones. Furthermore, the paper introduces a convergence analysis from a theoretical perspective.

**Strengths:**

The paper has a clear problem framing about robust few-shot adaptation of CLIP with PEFT, filling a gap left by prompt-only approaches.

The method is simple to implement and explained with theoretical analyses.

The proposed method obtains strong gains in robustnss at low shots, with competitive clean accuracy.

Theoretical analysis gives convergence to a stationary point.

**Weaknesses:**

Robustness is evaluated mainly against PGD in L-inifinty norm. There are also some other cases like L2-robustness and AutoAttack Robustness Evaluation.

It seems that at higher shots, the robustness advantage under PGD narrows and sometimes trails the best prompt baseline on clean accuracy.

Experiments focus on ViT-B/16 and ViT-B/32; larger or newer CLIP/SigLIP backbones are not explored.

The convergence guarantee relies on smoothness/boundedness assumptions.

**Questions:**

What is the exact trainable-parameter footprint versus full adversarial fine-tuning, and how does computational cost scale with the hyperparameter?

Can adversarially updating prompts or text-encoder inputs further improve robustness given that both encoders receive LoRA adapters?

How sensitive are results to the PGD step schedule and budget?

Would stronger inner maximizers (adversary generation) (e.g., MI-FGSM/AutoAttack variants) change conclusions?

---

### Note · Authors · 2025-12-11

I have read and agree with the venue's withdrawal policy on behalf of myself and my co-authors.